# KNOWLEDGE-GROUNDED REINFORCEMENT LEARNING

## ABSTRACT

Receiving knowledge, abiding by laws, and being aware of regulations are common behaviors in human society. Bearing in mind that reinforcement learning (RL) algorithms benefit from mimicking humanity, in this work, we propose that an RL agent can act on external guidance in both its learning process and model deployment, making the agent more socially acceptable. We introduce the concept, Knowledge-Grounded RL (KGRL), with a formal definition that an agent learns to follow external guidelines and develop its own policy. Moving towards the goal of KGRL, we propose a novel actor model with an embedding-based attention mechanism that can attend to either a learnable internal policy or external knowledge. The proposed method is orthogonal to training algorithms, and the external knowledge can be flexibly recomposed, rearranged, and reused in both training and inference stages. Through experiments on tasks with discrete and continuous action space, our KGRL agent is shown to be more sample efficient and generalizable, and it has flexibly rearrangeable knowledge embeddings and interpretable behaviors.

## 1 INTRODUCTION

Incorporating external guidance into learning is a commonly seen behavior among humans. We can speed up our learning process by referring to useful suggestions. At the same time, we beware of external regulations for safe and ethical reasons. On top of following external guidance, we humans can learn our own strategies to complete a task. We can also arbitrarily recompose, rearrange, and reuse those strategies and external guidelines to solve a new task and adapt to environmental changes. Imitating human behaviors has been shown to benefit reinforcement learning (RL) (Billard et al., 2016; Sutton and Barto, 2018; Zhang et al., 2019). However, how an RL agent can achieve the above capabilities remains challenging.

Different approaches have been proposed to develop some of these abilities. One branch of previous work in RL has studied how an agent can learn from demonstrations provided externally as examples of completing a task (Ross et al., 2011; Rajeswaran et al., 2017; Duan et al., 2017; Nair et al., 2018; Goecks et al., 2019; Ding et al., 2019). Another branch of previous research has investigated how an agent can learn reusable policies. This branch of research includes (1) transferring knowledge among tasks with similar difficulty (Parisotto et al., 2015; Yin and Pan, 2017; Gupta et al., 2018a; Liu et al., 2019; Tao et al., 2021) and (2) learning multiple reusable skills to solve a complex task in a divide-and-conquer manner (Bacon et al., 2017; Frans et al., 2017; Nachum et al., 2018a; Eysenbach et al., 2018; Kim et al., 2021; Tseng et al., 2021). These methods allow an agent to learn a new task with fewer training samples. However, demonstrations are too task-specific to be reused in a new task, and incorporating external guidelines from different sources into existing knowledge-reuse frameworks is not straightforward. Moreover, current learning-from-demonstration and knowledge-reuse approaches lack the flexibility to rearrange and recompose different demonstrations or knowledge, so they cannot dynamically adapt to environmental changes.

In this work, we introduce Knowledge Grounded Reinforcement Learning (KGRL), a novel problem with the following goal: *An RL agent can learn its own policy (knowledge) while referring to external knowledge. Meanwhile, all knowledge can be arbitrarily recomposed, rearranged, and reused anytime in the learning and inference stages.* A KGRL problem simultaneously considers the following three questions: (1) How can an agent follow a set of external knowledge from different sources? (2) How can an agent efficiently learn new knowledge by referring to external ones? (3) What is a proper

representation of knowledge such that an agent can dynamically recompose, rearrange, and reuse knowledge in both the learning process and model deployment?

To address the challenges in KGRL, we propose a simple yet effective actor model with an embedding-based attention mechanism. This model is orthogonal to training algorithms and easy to implement. In this model, each external-knowledge policy is paired with one learnable embedding. An internal-knowledge policy, which the agent learns by itself, is also paired with one learnable embedding. Then another learnable query embedding performs an attention mechanism by attending to each knowledge embedding and then decides which knowledge to follow. With this attention mechanism, the agent can learn new skills by following a set of external guidance. In addition, the knowledge and query embeddings design disentangles each knowledge from the selection mechanism, so a set of knowledge can be dynamically recomposed or rearranged. Finally, all knowledge policies are encoded into a joint embedding space, so they can be reused once their embeddings are appropriately learned.

We evaluate our proposed KGRL method for grid navigation and robotic manipulation tasks. The results demonstrate that our method answers all the questions considered in KGRL. In analyses, we show that the proposed approach achieves sample efficiency, generalizability, compositionality, and incrementality, which are the four essential components of efficient learning (Kaelbling, 2020). At the same time, our method also enables behavior interpretability because of the embedding-based attention mechanism.

Our contributions are:

- We introduce KGRL, a novel RL problem addressing how an agent can repeatedly refer to an arbitrary set of external knowledge for learning new skills.
- We propose a novel actor model that achieves the goals of KGRL and is orthogonal to training algorithms.
- We demonstrate in the experiments that our KGRL method satisfies the four properties of efficient learning (Kaelbling, 2020) and learns interpretable strategies.

## 2 RELATED WORK

Several lines of research in RL have studied how an agent can incorporate external demonstrations/regulations into learning or learn reusable policies. Here we summarize the differences between their research goals and the ones in the KGRL formulation.

**Learning from demonstrations/external strategies.** Providing expert demonstrations or external strategies to an RL agent is a popular way to incorporate external guidance into learning. Demonstrations are (sub-)optimal examples of completing a task and represented as state-action pairs; external strategies are (sub-)optimal policies that are represented in a specific form, e.g., fuzzy logic. Previous work has leveraged demonstrations or external strategies by introducing them into the policy-update steps of RL (Hester et al., 2017; Rajeswaran et al., 2017; Vecerik et al., 2017; Nair et al., 2018; Pfeiffer et al., 2018; Goecks et al., 2019; Kartal et al., 2019; Zhang et al., 2020; 2021) or performing imitation learning (IL) (Abbeel and Ng, 2004; Ross et al., 2011; Ho and Ermon, 2016; Duan et al., 2017; Peng et al., 2018a;b; Ding et al., 2019). Learning from demonstrations (LfD) and IL require sufficient number of optimal demonstrations, which are difficult to collect, to achieve sample-efficient learning. Moreover, those demonstrations are task-dependent and can hardly be reused in other tasks. Compared to LfD and IL, learning from external strategies, including KGRL, can achieve efficient learning by referring to highly imperfect knowledge. Also, KGRL can incorporate general knowledge that is not necessary task-related. Finally, our KGRL approach learns a uniform representation of knowledge in different forms, so it supports flexible recomposition/rearrangement/reuse of an arbitrary set of knowledge. These abilities are not all achievable by previous work in this direction.

**Safe RL.** Other than demonstrations, regulations are also essential guidance that can help an agent achieve effective learning. Safe RL, which incorporates safety regulations (constraints) into learning and inference stages, has gained more attention over the past decade. Current research in safe RL mainly includes safety in learning through (1) jointly optimizing the expected return and safe-related cost (Ammar et al., 2015; Achiam et al., 2017; Chow et al., 2018; Stooke et al., 2020; Ding et al.,

2021) and (2) restricting exploration in a safe space (Garcia and Fernández, 2012; Berkenkamp et al., 2017; Alshiekh et al., 2018; Cheng et al., 2019; Thananjeyan et al., 2021; Liu et al., 2022). The constraints in safe RL can be softly or strictly imposed, where the former aims to decrease the number of safety violations, and the latter prohibits any of them in both learning and inference stages. Compared to KGRL, safe RL usually does not consider reusing the safety constraints in a new task or dynamically adjusting regulations according to the environmental changes.

**Knowledge transfer and meta-RL.** How an agent can transfer previously learned knowledge to a new task is a well-recognized challenge. Several studies have focused on transferring the knowledge learned in a single task to a multi-task policy (Parisotto et al., 2015; Rusu et al., 2015; Yin and Pan, 2017; Xu et al., 2020; Tao et al., 2021). On the other hand, a line of research in meta-RL investigates how to quickly adapt the knowledge learned in multiple tasks to a new one (Finn et al., 2017; Gupta et al., 2018a;b; Nagabandi et al., 2018; Liu et al., 2019). These two research directions do not enable incorporating external guidance from different sources into learning, so their transferred knowledge is limited to the one learned among similar environments (Glatt et al., 2016). In addition, they require extensive amount of training data before a set of knowledge can be reused in a new task. Compared to previous work in knowledge transfer and meta-RL, KGRL allows an agent to follow and reuse an arbitrary set of knowledge obtained from different sources in a new task, and these sources can be very different from the new task. Moreover, our KGRL approach does not require a separate process, such as policy distillation (Rusu et al., 2015), or numerous training samples collected from multiple environments to learn transferable knowledge.

**Hierarchical RL (HRL).** Other than transferring knowledge among tasks with similar difficulty, another popular line of research, HRL, aims to decompose a complex task into a hierarchy of subtasks and learn a reusable skill for each subtask. Previous research in HRL has focused on learning a hierarchy of a complex task (Dayan and Hinton, 1992; Sutton et al., 1999; Kulkarni et al., 2016; Levy et al., 2017; Nachum et al., 2018b; Jiang et al., 2019), discovering options/subtasks/subgoals (Stolle and Precup, 2002; Bacon et al., 2017; Vezhnevets et al., 2017; Nachum et al., 2018a; Riemer et al., 2018; Eysenbach et al., 2018; Khetarpal and Precup, 2019; Zhang and Whiteson, 2019; Kim et al., 2021), and transferring learned skills among different tasks (Konidaris and Barto, 2006; Frans et al., 2017; Tessler et al., 2017; Peng et al., 2019; Qureshi et al., 2019; Tseng et al., 2021). Compared to the skills in HRL, knowledge in KGRL can be partially contained in, be more complex than, or even has little to do with the task. That is, it is not limited to a skill that solves a subtask. This property enables flexibility of providing knowledge and improves exploration in an unseen environment. In addition, a skill in HRL is usually assigned with a fixed index or a task-dependant subgoal, making an agent difficult to dynamically recompose or rearrange skills in learning or inference stages. On the other hand, our KGRL method disentangles knowledge representation from the high-level selection mechanism of a task, making an agent able to dynamically adapt to environmental changes anytime.

**Program-guided RL.** Similar to HRL, program-guided RL also decomposes a complex task into subtasks and learns subpolicies for them. However, instead of learning a hierarchy of the task, program-guided RL guides an agent to complete a task with a program, which explicitly specifies the flow of executing the subtasks given the environmental conditions (Sun et al., 2019; Brooks et al., 2021; Trivedi et al., 2021; Yang et al., 2021; Zhao et al., 2021). This guidance requires a structured and unambiguous hierarchy of a task and is more task-dependent compared to knowledge in KGRL. Moreover, to reuse a previously learned subpolicy, program-guided RL needs to first generate a new program with a new control flow, hence lacking the flexibility to dynamically recompose and rearrange subpolicies.

## 3  KGRL PROBLEM FORMULATION

In this section, we formally define the problem formulation of *Knowledge-Grounded Reinforcement Learning (KGRL)*. We consider the family of discrete-time finite-horizon Markov Decision Processes (MDPs) with knowledge guidance for an environment. Such an MDP is formulated as a tuple $(\mathcal{S}, \mathcal{A}, \mathcal{T}, \mathcal{G}, \mathcal{R}, \rho, \gamma)$, where $\mathcal{S}$ is the state space, $\mathcal{A}$ is the action space, $\mathcal{T} : \mathcal{S} \times \mathcal{A} \times \mathcal{S} \to \mathbb{R}$ is the transition probability distribution, $\mathcal{G}$ is the *knowledge set*, $\mathcal{R}$ is the reward function, $\rho$ is the initial state distribution, and $\gamma$ is the discount factor.

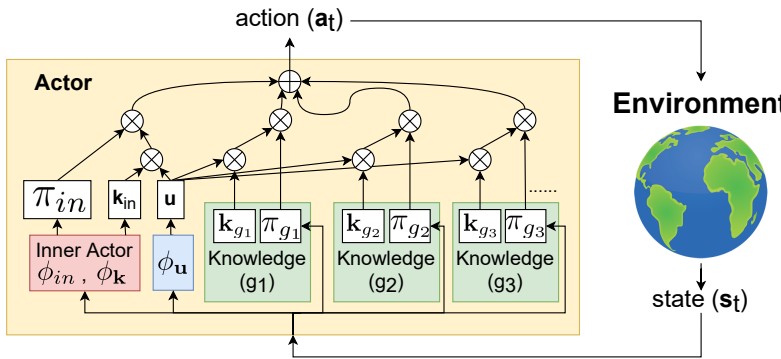

Figure 1: The proposed model architecture of a KGRL agent.

Each element in the knowledge set, $g \in \mathcal{G}$, can be general knowledge, such as commonsense, suggestion, guideline, principle, specification, rule, or restriction, described as a knowledge mapping $\pi_g : \mathcal{S} \rightarrow \mathcal{A}$. This mapping represents the low-level execution strategy of each knowledge and can be in any form, such as a hand-crafted rule, fuzzy logic, or a neural network, as long as it outputs an action given a state. In our experiments, we focus on $g$ being suggestions in the form of hand-crafted rules or neural networks. This is because flexible recomposition/rearrangement of a set of suggestions is fundamental to efficient learning but is currently not achievable by existing methods. Moving forward, this setup can lay a foundation to incorporate a more extensive knowledge set.

The goal of a KGRL agent is to maximize the expected rewards and develop the following attributes: (1) The agent learns its own knowledge by referring to external one. (2) The agent can recompose/re-arrange/reuse knowledge in training and inference stages as well as new environments. (3) The agent's behaviors are interpretable in terms of which knowledge it refers to at each time step.

## 4 PROPOSED APPROACH

We propose a novel actor model for KGRL with the concept inspired by the self-attention mechanism (Vaswani et al., 2017). The actor model includes an inner actor and a set of external knowledge mappings, $\{\pi_{g_1}, \pi_{g_2}, \ldots, \pi_{g_n}\}$. The actor architecture is illustrated in Figure 1. The idea behind this design is that the actor will decide to follow an external guidance or its own policy given a state $\mathbf{s}_t \in \mathcal{S}$ at the $t$-th time step. This dynamic decision making should eventually lead the agent to receive higher accumulated rewards based on the reward function $\mathcal{R}$. Moreover, for the external knowledge set to be arbitrarily recomposed/rearranged and deployed without further training, each knowledge should be represented in a form that is order-invariant, and the knowledge-selection mechanism should be independent from which set of knowledge is used during training. In the following sections, we introduce the detailed design of each component in the proposed actor model.

### 4.1 ACTOR MODEL ARCHITECTURE

**External Knowledge.** An important property of a KGRL policy is the flexibility of knowledge recomposition/rearrangement/reusability during training and testing phases. To obtain this flexibility, we propose a uniform representation for each knowledge so that the components of a knowledge set is order-invariant. We assign a learnable *key* (embedding) for each external knowledge mapping, which is a $d_k$-dimensional vector denoted as $\mathbf{k}_{g_j} \in \mathbb{R}^{d_k}$, for $j = 1, 2, \ldots, n$. Note that the knowledge key $\mathbf{k}_{g_j}$ represents the *entire* knowledge mapping $\pi_{g_j}$ instead of each state-action pair suggested by this mapping, so $\mathbf{k}_{g_j}$ does not depend on states or actions. Since each knowledge is encoded as an *independent* key, which lies in a joint embedding space, one can change the number of knowledge by adding/removing keys to/from the space to adapt to the dynamical changes in an environment. Meanwhile, one can also combine knowledge keys from different tasks in a new learning/evaluation process.

**Inner Actor.** An external knowledge set may not contain policies that can directly solve a specific task, so an agent needs to be capable of developing its own strategies. Therefore, we add an *inner actor*

in the actor model of KGRL. An inner actor contains two learnable function approximators $\phi_{in}$ and $\phi_{\mathbf{k}}$. Given a state $\mathbf{s}_t$, the function approximator $\phi_{in}$ predicts an inner policy, $\pi_{in} = \phi_{in}(\mathbf{s}_t) : \mathcal{S} \rightarrow \mathcal{A}$, and $\phi_{\mathbf{k}}$ generates an *inner key*, $\mathbf{k}_{in} = \phi_{\mathbf{k}}(\mathbf{s}_t) \in \mathbb{R}^{d_k}$. The inner policy serves the same purpose as a policy in regular RL, which represents the strategy that an agent learns by itself through interactions with the environment. Similar to an external knowledge mapping $\pi_{g_i}$, the inner policy $\pi_{in}$ can be viewed as the *internal knowledge mapping* with a paired learnable inner key $\mathbf{k}_{in}$. Note that unlike an external knowledge key, the inner key varies according to a given state, because the inner strategy will change as the agent learns from more experiences over time.

**Query.** Given the external and internal knowledge mappings, an agent needs to decide which knowledge to refer to. Importantly, this selection mechanism should be disentangled from a knowledge set to support its flexible recomposition/rearrangement ability. That is, an agent should still be able to select a proper knowledge mapping when a different knowledge set is provided. To achieve this goal, we use an *attention mechanism* to query knowledge. This attention mechanism is implemented by adding another learnable function approximator $\phi_{\mathbf{u}}$ that generates a *query* vector, denoted as $\mathbf{u} = \phi_{\mathbf{u}}(\mathbf{s}_t) \in \mathbf{R}^{d_k}$, given a state $\mathbf{s}_t$. The query will be attended to the keys to calculate the weights for each knowledge mapping. Note that the proposed model predicts the query and keys separately instead of directly predicting the weight for each guidance. In this way, we disentangle the selection component (the query) from the knowledge set (the keys). The following section describes how we compute the weights.

## 4.2 ACTION PREDICTION

**Attentions.** We compute the attention of the query $\mathbf{u}$ over the inner keys and external knowledge keys by their dot products. The results become the weights for the inner and external guidance:

$$w_{in} = \mathbf{u} \cdot \mathbf{k}_{in} ,$$
$$w_{g_j} = \mathbf{u} \cdot \mathbf{k}_{g_j} . \tag{1}$$

If $w_{in}$ is larger, then the agent is more likely to adopt the inner policy. On the other hand, if $w_{g_j}$ is larger, then the agent is more likely to follow the guidance from the $j$-th external knowledge.

**Final Policy.** A straightforward way to describe the final policy given the weights $w_{in}$ and $w_{g_j}$ and their corresponding knowledge mappings $\pi_{in}$ and $\pi_{g_j}$, for $j = 1, 2, \ldots, n$, is

$$\mathbf{a}_t \sim \pi(\cdot|\mathbf{s}_t), \tag{2}$$

$$\pi(\cdot|\mathbf{s}_t) = w_{in}\pi_{in}(\cdot|\mathbf{s}_t) + \sum_{j=1}^{n} w_{g_j}\pi_{g_j}(\cdot|\mathbf{s}_t), \tag{3}$$

which is a weighted sum of each strategy. We apply equation (3) to environments with a discrete action space, where each knowledge mapping predicts a probability vector. However, we found that in environments with a continuous action space, taking actions sampled from equation (3) will guide an agent to complete a task less efficiently.

We explain by a simple example why computing a weighted sum is a less favorable choice when integrating multiple knowledge mappings in a continuous action space. Figure 2 shows a 2D environment where an agent navigates to avoid the obstacle and reach the goal. The agent is provided with two knowledge mappings. Their suggested actions at the current time step are plotted as an arrow in orange and blue respectively, and their weights are both $0.5$. The weighted sum of these two action candidates is plotted as a green arrow, which guides the agent to go directly into the obstacle. This example demonstrates that a weighted sum of multiple actions in a continuous action space can lose critical information contained in individual actions and lead to worse behaviors. Hence, we instead sample only one action candidate according to the weights.

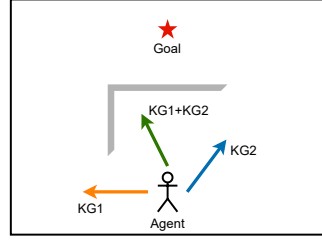

Figure 2: An example of incorporating multiple actions by a weighted sum. It shows that a weighted sum of actions can lead the agent to go into the obstacle (the gray wall).

To sample an action suggested by one of the knowledge mappings while making the entire actor model differentiable, we first sample an

---

**Algorithm 1:** Knowledge-Guided Reinforcement Learning (KGRL) with Our Actor Model

---

**Input:** environment $\mathcal{E}$ with MDP $(\mathcal{S}, \mathcal{A}, \mathcal{T}, \mathcal{G}, \mathcal{R}, \rho, \gamma)$
**Input:** external knowledge mappings $\pi_{g_1}, \pi_{g_2}, ..., \pi_{g_n}$
initialize $\mathbf{k}_{g_j}$ for $j = 1, 2, ..., n$
initialize $\phi_{in}, \phi_{\mathbf{k}}, \phi_u$
**for** *each time step* **do**
   | Generate $\mathbf{k}_{g_1}, \ldots, \mathbf{k}_{g_n}, \mathbf{k}_{in}, \pi_{in}, \mathbf{u}$
   | Compute $w_{in}, w_{g_1}, \ldots, w_{g_n}$ according to equation (1)
   | Sample an action $\mathbf{a}_t \sim \pi(\cdot|\mathbf{s}_t)$ according to equations (2)
   | Apply $\mathbf{a}_t$ and observe a new state and reward $\mathbf{s}_{t+1}, r_t$ from $\mathcal{E}$
   | Update $\phi_{in}, \phi_k, \phi_u, \mathbf{k}_{g_1}, \ldots, \mathbf{k}_{g_n}$ by any RL algorithm
return $\phi_{in}, \phi_{\mathbf{k}}, \phi_u, \mathbf{k}_{g_1}, \ldots, \mathbf{k}_{g_n}$

---

element from the set $\{in, g_1, g_2, \ldots, g_n\}$ according to the weights $\{w_{in}, w_{g_1}, w_{g_2}, \ldots, w_{g_n}\}$ using Gumbel softmax (Jang et al., 2016):

$$e \sim \texttt{gumbel\_softmax}(w_{in}, w_{g_1}, w_{g_2}, ..., w_{g_n}). \quad (4)$$

Then given a state $\mathbf{s}_t$, an action is sampled from the knowledge mapping $\pi_e(\cdot|\mathbf{s})$ using the reparameterization trick, where $\phi_e$ can be one of $\{\phi_{in}, \phi_{g_1}, \ldots, \phi_{g_n}\}$ according to the sampled $e$.

### 4.3 LEARNING ALGORITHMS

The proposed actor model can be end-to-end learned with any existing RL algorithm, and the implementation is simple. Algorithm 1 describes our proposed KGRL algorithm.

## 5 EXPERIMENTS

We validate the proposed KGRL actor model on two sets of widely used environments with different properties: MiniGrid (Chevalier-Boisvert et al., 2018) and OpenAI-Robotics (Plappert et al., 2018). Through experiments on these environments, we answer the following four questions: (1) Can our KGRL agent learn effective new knowledge by referring to an arbitrary set of external guidance? (2) Can the knowledge policies and their learned embeddings be recomposed, rearranged, and reused to solve a new task? (3) Is our KGRL method sample efficient and generalizable? (4) Does our KGRL agent learn interpretable behaviors? The implementation details can be found in Appendix.

**Baselines.** We compare our KGRL method with the following baselines that incorporate external guidance in different ways. (1) **Behavior cloning (BC)**: An agent follows only the external knowledge to solve a task. The results of this strategy can be seen as the upper bound of BC, where the demonstration data is generated by applying the external knowledge. (2) **RL**: An agent learns a policy from scratch by RL. No external guidance is incorporated. (3) **RL+BC**: An agent learns a policy by RL with BC signals provided from external knowledge. We follow the method proposed in Nair et al. (2018) to incorporate BC losses into RL.

### 5.1 MINIGRID

**Environments.** We evaluate all methods in the following MiniGrid environments with discrete action space and sparse rewards: Empty-5x5, Empty-Random-5x5, Empty-16x16, Unlock, DoorKey-5x5, and DoorKey-8x8. Among them, DoorKey are the environments that an agent needs to pick up a key, open a door, and then reach a goal, which can use the skills learned in Unlock (pick up a key and open a door) and Empty (reach a goal). We choose these environments to study if our method can recompose/reuse previously learned knowledge. The observation space of these environments is the directed first person view with a range of 5x5 grid, and the action space has seven discrete actions.

**Setup.** We compare BC and Proximal Policy Optimization algorithm (Schulman et al., 2017) under RL (PPO), RL+BC (PPO-BC), and KGRL (PPO-KGRL) formulations. In PPO-KGRL, we use the

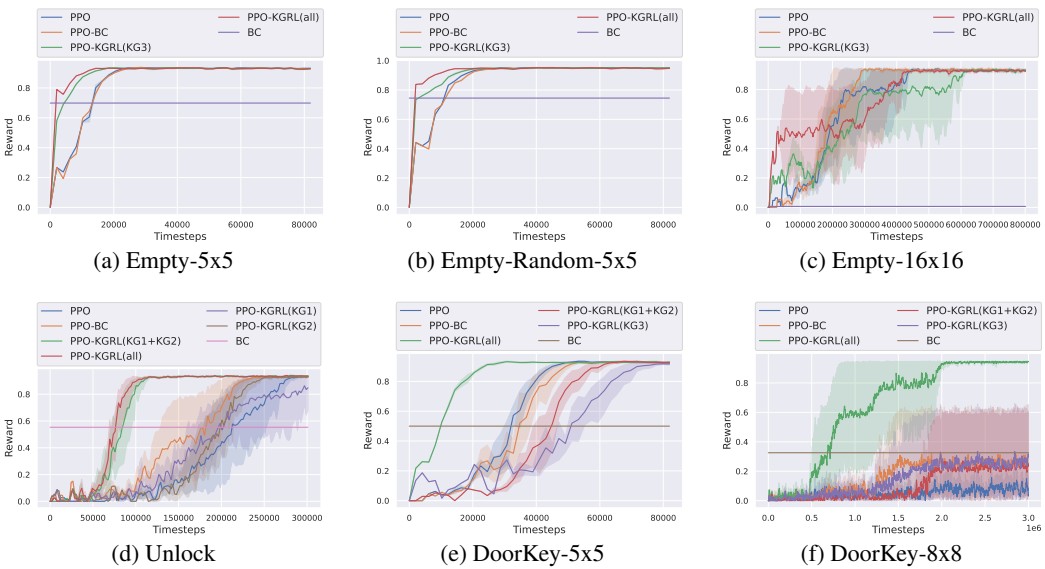

(a) Empty-5x5      (b) Empty-Random-5x5      (c) Empty-16x16

(d) Unlock      (e) DoorKey-5x5      (f) DoorKey-8x8

Figure 3: The learning curves of different agents in MiniGrid environments. Our KGRL method achieves better sample efficiency in most of the environments. Also, adding redundant knowledge in the proposed model does not affect the performance.

inner actor model same as the actor model of PPO and add the learnable inner key, knowledge keys, and query. We define three simple knowledge policies for the MiniGrid environments. **(KG1)** pick up a key: if there is a key in the observation, "go in the direction" of the key; if the key is just in front of the agent, pick up the key. **(KG2)** open a door: if there is a door in the observation, "go in the direction" of the door; if the door is just in front of the agent, open the door. **(KG3)** reach a goal: if there is a goal in the observation, "go in the direction" of the goal.

**Results.** Figure 3 shows the training results of the baseline methods and PPO-KGRL with different knowledge combinations. In most of the environments, PPO-KGRL achieves significantly better sample efficiency given sub-optimal knowledge. Also in general, the improvement is larger when the task is more complex (Empty < Unlock < DoorKey). Note that PPO does not show any success in DoorKey-8x8 within three million steps. However, we found that in Empty-16x16, our KGRL agent learns more quickly at the beginning but slows down its progress over time. The reason might be that Empty-16x16 requires only exploration instead of other skills, so the agent cannot benefit too much by following external guidance. Moreover, in a small environment such as DoorKey-5x5, the advantages of providing external knowledge is difficult to demonstrate due to extra learnable parameters in our actor model.

We also study the effects of different knowledge combinations for PPO-KGRL. In Unlock, we found that KG2 is more helpful than KG1; in DoorKey-8x8, KG3 is more helpful than KG1+KG2. We conjecture that the knowledge which directly helps receive the rewards guides an agent more effectively in these environments. Also, as shown in Figure 3(a-d), adding redundant knowledge will not affect the performance. This observation demonstrates the benefit of disentangling the knowledge embeddings and query, thus increasing the flexibility of providing external guidance.

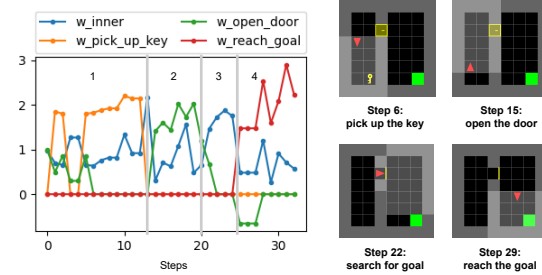

Figure 4: The interpretation of the agent's behaviors. The left image shows the (unnormalized) weights for each knowledge.

**Policy Interpretation.** Figure 4 shows that our KGRL method learns interpretable agent behaviors. The weight changes indicate that in a DoorKey-8x8 environment, the agent chooses to sequentially "pick up the key", "open the door", "search for the goal", and "reach the goal". Note that the attention to the inner policy does not disappear throughout the entire episode. This shows that the inner policy helps an agent explore the environment.

**Analysis of Generalizability.** To test the generalizability of PPO and PPO-KGRL, we train an agent in an empty environment with a fixed goal (Empty) and evaluate the agent in an empty environment with a randomized goal (Empty-Random). This experimental design verifies whether the agent memorizes the goal position or really understands how to find the goal. As shown in the left column of Table 1, the returns of PPO-KGRL achieve an average of 0.94 and a minimum of 0.77. This is much closer to the upper bound, which is the average return of a PPO-KGRL agent trained on the Empty-Random environment.

| Init. Env. | Empty | DoorKey-5x5 | | | DoorKey-8x8 |
|---|---|---|---|---|---|
| Test Env. | Empty-Random | Empty | Empty-Random | Unlock | Unlock |
| PPO | 0.92 (0.71) | 0.55 (0.05) | 0.79 (0.03) | 0.43 (0) | 0 (0) |
| PPO-KGRL | **0.94 (0.77)** | **0.88 (0.55)** | **0.90 (0.55)** | **0.54 (0)** | **0.85 (0)** |
| Upper Bound | 0.95 (0.84) | 0.93 (0.87) | 0.95 (0.84) | 0.94 (0.84) | 0.94 (0.84) |

Table 1: Zero-shot evaluations on three environments (Empty, Empty-Random, Unlock) of RL agents trained on an initial environment (Empty and DoorKey) using PPO or PPO-KGRL algorithms. The average and minimal returns are shown for each evaluation. Our KGRL method is more generalizable and effective in transferring previously learned knowledge.

**Analysis of Reusing, Recomposing, and Rearranging Knowledge.** For our KGRL method, we verify if the external and internal knowledge policies and their embeddings can be reused by the following two sets of experiments: (1) zero-shot transferring the knowledge policies and embeddings to a simpler task and (2) initializing the external guidance by those knowledge policies and embeddings when learning a more complex task. The second set of experiments also verifies if the knowledge policies and embeddings can be recomposed and rearranged.

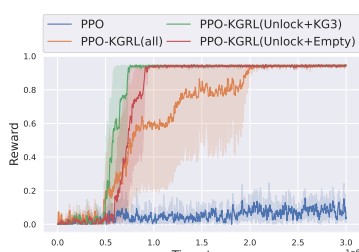

In the first set of experiments, we zero-shot evaluate an agent trained in DoorKey environments on Empty, Empty-Random, and Unlock environments. The four columns on the right side of Table 1 show the average and minimal returns. We found that an agent trained with PPO-KGRL significantly outperforms the one trained with PPO in all environments. These results demonstrate that our KGRL policies have better reusability.

Figure 5: The knowledge reusability experiments on MiniGrid-DoorKey-8x8. Our KGRL method further improve sample efficiency by reusing previously learned knowledge.

In the second set of experiments, we train three PPO-KGRL agents with external knowledge being:

1. all: KG1 + KG2 + KG3. This set of knowledge is the same one used in Figure 3.

2. Unlock (reused) + KG3: the inner knowledge policy and embeddings learned in Unlock + KG3. This set of knowledge recomposes and rearranges the first set of knowledge.

3. Unlock (reused) + Empty (reused): the inner knowledge policy and embeddings learned in Unlock + the ones learned in Empty-Random. This set of knowledge recomposes and rearranges the second set of knowledge.

Figure 5 shows the training results of PPO and PPO-KGRL with different sets of external knowledge. The two KGRL agents that reuse previously learned knowledge perform better than PPO and PPO-KGRL(all). Also surprisingly, we found that the number of samples required by PPO-KGRL that reuses the Unlock knowledge *plus* the one required by learning the Unlock knowledge is even less than the one required by PPO-KGRL(all), which learns with KG1+KG2+KG3. These results indicate

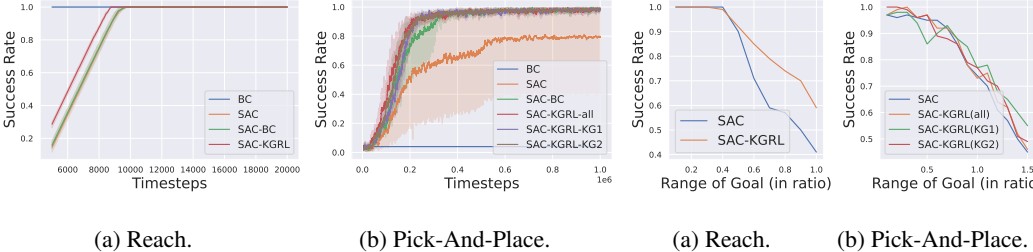

(a) Reach.  (b) Pick-And-Place.  (a) Reach.  (b) Pick-And-Place.

Figure 6: The learning curves of different agents in OpenAI robotic environments. Our KGRL method leads to better sample efficiency and stability in both environments.

Figure 7: Results of the policy trained on 0.1x the original goal range. Our KGRL method is less affected by the changes in the environment.

that with the abilities to flexibly reuse, recompose, and rearrange knowledge, our KGRL method can further enhance the training efficiency.

## 5.2 ROBOTICS

**Environments.** We also verify the proposed KGRL approach in the following OpenAI robotic environments with a continuous action space and sparse rewards: Reach and Pick-And-Place. Both environments contain a task of reaching a randomly sampled goal. Similar to the MiniGrid environments, these two tasks can demonstrate the ability of our KGRL method reusing, recomposing, and rearranging previously learned knowledge when solving a new task. An observation of these environments includes the position and velocity of the end-effector and object (if it exists) and the position of the goal. An action of these environments contains the position variation of the end-effector and the distance between the two fingers.

**Setup.** We compare BC and the soft actor-critic (Haarnoja et al., 2018) algorithm with hindsight experience replay (Andrychowicz et al., 2017) under RL (SAC), RL+BC (SAC-BC), and KGRL (SAC-KGRL) formulations. In SAC-KGRL, we add the learnable inner key, knowledge keys, and query on top of the actor model of SAC. We define the following two external knowledge policies for the OpenAI robotic environments, where $\mathbf{p}_{ee} \in \mathbb{R}^3$ is the position of the end-effector and $\mathbf{p}_{obj} \in \mathbb{R}^3$ is the position of the object. (**KG1**) move straightly to the goal: if $\|\mathbf{p}_{ee} - \mathbf{p}_{obj}\|_2 < \varepsilon$, move straightly to the goal with the gripper closed; otherwise, stay unmoved. Note that there is no object in Reach, so $\varepsilon$ is set to infinity in this environment, and **KG1** always guides the agent towards the goal. (**KG2, for Pick-And-Place only**) move straightly to the object: if $\|\mathbf{p}_{ee} - \mathbf{p}_{obj}\|_2 \geq \varepsilon$, move straightly to the object with the gripper opened; otherwise, stay unmoved.

**Results.** Figure 6 shows the training results of Reach and Pick-And-Place. In both environments, SAC-KGRL requires less samples to succeed and shows more stability. Compared to SAC-BC, our method better utilizes external guidance regardless of its quality. Moreover, the success of SAC-KGRL in Pick-And-Place demonstrates that our KGRL agent can learn an effective strategy given sub-optimal knowledge. We also study the effects of different knowledge combinations in Pick-And-Place. The learning curves indicate that using both KG1 and KG2 leads to slightly better sample efficiency than using only one of them, but there is no significant difference among the three combinations.

**Analysis of Generalizability.** We analyze if our KGRL policies are more generalizable by training in both environments with a smaller goal range and testing with larger goal ranges. The training goal range is 0.1 times smaller than the original one, and the testing goal ranges are the original one multiplied by 0.1 to 1. Figure 7 shows the generalizability results. The success rate of SAC and SAC-KGRL agents both decreases as the testing goal range increases, but the SAC-KGRL agent in Reach and the SAC-KGRL(all) agent in Pick-And-Place are less affected by the changes, especially when the goal ranges are larger. These results suggest that an KGRL agent has more potential to generalize to unseen environments.

| Environment | Reach | | | Pick-And-Place | | | |
|---|---|---|---|---|---|---|---|---|
| Knowledge | All | IA | KG1 | All | IA + KG1 | IA + KG2 | IA | KG1 + KG2 |
| Success Rate | 1 | 1 | 1 | 0.99 | 0.99 | 0.99 | 0.99 | 0.05 |

Table 2: Evaluation results of removing part of the knowledge set of our KGRL policies (IA: inner actor, All: IA + all KGs). Our proposed method enables the inner actor to *mimic* the strategies of external knowledge, and it may become a more powerful piece of new knowledge.

**Analysis of Knowledge Control Flexibility.** We demonstrate the flexibility of our method re-composing and rearranging knowledge by removing part of the knowledge set during evaluation. We evaluate Reach with the policies learned by SAC-KGRL and Pick-And-Place with the policies learned by SAC-KGRL(all). Table 2 shows the testing results of both tasks after removing part of the knowledge set. The learned inner actors in both environments achieve high success rate without the external guidance, indicating that a KGRL agent learns to not only *follow* the guidance but also *mimic* its strategy. This imitation process allows an agent to develop an inner policy that outperforms external strategies, and itself can then become new knowledge useful for solving other tasks.

| Model | SAC | SAC-KGRL-all | SAC-KGRL-KG1 | SAC-KGRL-KG2 |
|---|---|---|---|---|
| 100k Training Steps | 0.24 | 0.56 | 0.25 | 0.48 |
| 200k Training Steps | 0.61 | 0.99 | 0.99 | 0.98 |
| Best | 0.8 | 1 | 1 | 1 |

Table 3: Zero-shot transfer results of Pick-And-Place policies to Reach. Our proposed method is more effective and efficient in reusing learned knowledge.

**Analysis of Knowledge Reusability.** To demonstrate that our KGRL method enables better knowl-edge reusability, we apply zero-shot transfer from the learned Pick-And-Place policies to Reach. Note that the observation spaces of Reach and Pick-And-Place are different since there is no object in Reach. Hence, we apply zeros as the object's relative position and velocity in Reach to unify their observation spaces. Table 3 shows the success rate of different Pick-And-Place policies applied to Reach. The results indicate that SAC-KGRL is more efficient in reusing knowledge since the transfers are mostly successful after only 200k training steps.

## 6 CONCLUSION AND DISCUSSION

In this work, we introduce KGRL, a novel RL problem discussing how an agent can follow an arbitrary set of external guidance and learn its own policy. Also, the agent should be able to repeatedly reuse previously learned knowledge and dynamically recompose and rearrange knowledge to adapt to environmental changes. To address these challenges, we propose a simple yet effective actor model that performs an embedding-based attention mechanism. The experimental results show that the proposed model achieves the goal of KGRL. Our KGRL algorithm also satisfies the four properties of efficient learning (Kaelbling, 2020) and learns interpretable agent behaviors.

This work serves as an initial effort for the ultimate goal of KGRL, which is to automatically collect a large group of knowledge from different sources, such as books, the Internet, or previous training processes. This group of knowledge can be shared among multiple environments/tasks and continuously expanded, and a wide variety of artificial agents can use it as their guidance. Moving forward, the following research directions in KGRL are worth exploring: (1) incorporating restrictions that should be strictly followed during the learning and inference stages, (2) composing knowledge with different state and action spaces, and (3) dealing with more complex relationships among different knowledge, such as conditional dependence and conflicts. We hope this work can motivate future research in KGRL.

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

# A EXPERIMENTAL DETAILS

## A.1 MINIGRID ENVIRONMENTS

### A.1.1 ENVIRONMENTAL DETAILS

The environmental details can be found at `https://github.com/maximecb/gym-minigrid`.

### A.1.2 PREDEFINED KNOWLEDGE

KG1, KG2, and KG3 for MiniGrid environments are hand-crafted rules that output a probability vector given a state. We provide our code of defining the rules here.

```
1  agent_pos = (obs.image.shape[1]//2, obs.image.shape[2]-1)
2  expert_actions = torch.zeros((obs.image.shape[0], len(expert_rules), len(actions)), device='cuda:0')
3  img = obs.image[:,:,:,0]
4
5  def convert_pos_to_dir_actions(rule_id, start, goals, ids_to_remove=None):
6      img_id_meet_condition = goals[:,0][(start[0] < goals[:,1]).int().nonzero()]
7      expert_actions[img_id_meet_condition,rule_id,actions["Right"]] = 1
8      img_id_meet_condition = goals[:,0][(start[0] > goals[:,1]).int().nonzero()]
9      expert_actions[img_id_meet_condition,rule_id,actions["Left"]] = 1
10     img_id_meet_condition = goals[:,0][(start[1] > goals[:,2]).int().nonzero()]
11     expert_actions[img_id_meet_condition,rule_id,actions["Forward"]] = 1
12     if ids_to_remove is not None:
13         expert_actions[ids_to_remove,rule_id,actions["Right"]] = 0
14         expert_actions[ids_to_remove,rule_id,actions["Left"]] = 0
15         expert_actions[ids_to_remove,rule_id,actions["Forward"]] = 0
16
17 # KG1: pick up a key, id=5
18 if "get the key" in expert_rules:
19     rule_id = expert_rules.index("get the key")
20     key_pos = (img == 5).nonzero()
21     img_id_meet_condition = key_pos[:,0][((key_pos[:,1] == \
22         agent_pos[0]).int() * (key_pos[:,2] == agent_pos[1]-1).int()).nonzero()]
23     expert_actions[img_id_meet_condition,rule_id,actions["Pickup"]] = 1
24     convert_pos_to_dir_actions(rule_id, agent_pos, key_pos, ids_to_remove=img_id_meet_condition)
25
26 # KG2: open a door, id=4
27 if "open the door" in expert_rules:
28     rule_id = expert_rules.index("open the door")
29     door_pos = (img == 4).nonzero()
30     img_id_meet_condition = door_pos[:,0][((door_pos[:,1] == \
31         agent_pos[0]).int() * (door_pos[:,2] == agent_pos[1]-1).int()).nonzero()]
32     expert_actions[img_id_meet_condition,rule_id,actions["Toggle"]] = 1
33     convert_pos_to_dir_actions(rule_id, agent_pos, door_pos, ids_to_remove=img_id_meet_condition)
34
35 # KG3: reach a goal, id=8
36 if "go to the goal" in expert_rules:
37     rule_id = expert_rules.index("go to the goal")
38     goal_pos = (img == 8).nonzero()
39     convert_pos_to_dir_actions(rule_id, agent_pos, goal_pos)
```

### A.1.3 MODEL ARCHITECTURE

We use the same model architecture for all the six environments. Each external knowledge embedding is generated by the PyTorch Paszke et al. (2019) module, `nn.Embedding`, with $d_k = 8$. The inner actor network, $\phi_{in}$, is a 3-layer convolutional neural network followed by a 1-layer multi-layer perceptron (MLP). Its detailed architecture can be found in `https://github.com/lcswillems/rl-starter-files`. The inner key and query networks, $\phi_{\mathbf{k}}$ and $\phi_{\mathbf{u}}$, share the same MLP layer with $\phi_{in}$, and each of them is followed by Tanh activation and an output layer with the dimension being $d_k$.

### A.1.4 BASELINE ALGORITHMS

**BC.** Since KG1, KG2, and KG3 are not in conflict with one another, we can combine these three knowledge mappings (KG1+KG2+KG3). The results of BC is obtained by applying KG1+KG2+KG3 in all environments.

**PPO-BC.** The external knowledge policy KG1+KG2+KG3 can generate demonstrations to guide an agent. With this policy, we add the following BC loss to the objective of PPO:

$$L_{BC} = c_{BC} D_{KL}(\pi(\mathbf{s})||\pi_{KG}(\mathbf{s})), \tag{5}$$

where $c_{BC} \in \mathbb{R}$ is the coefficient of this loss, $D_{KL}(\cdot||\cdot)$ calculates the KL divergence, $\pi(\cdot)$ is the policy learned by an agent, and $\pi_{KG}(\cdot)$ is the external knowledge policy. We set $c_{BC} = 10^{-3}$ for all environments.

### A.1.5 HYPERPARAMETERS

Each training trial in Section 5.1 is run with 5 different random seeds, and we report the average performance of the 5 trained agents for analyses. For the underlying PPO algorithm, we follow the implementation of `https://github.com/lcswillems/rl-starter-files` with unchanged hyperparameters.

### A.2 OPENAI ROBOTIC ENVIRONMENTS

#### A.2.1 ENVIRONMENTAL DETAILS

A state $\mathbf{s}_t$ in unmodified Reach, where $\mathbf{s}_t \in \mathbb{R}^{10}$, contains the position and velocity of the end-effector, the distance between the two grippers, and their velocity. A state $\mathbf{s}_t$ in Pick-And-Place, where $\mathbf{s}_t \in \mathbb{R}^{25}$, contains the same information as that in Reach, the position, rotation, and velocity of the object, and the relative position between the object and the end-effector. In modified Reach, which is used in the analysis of knowledge reusability, a state $\mathbf{s}_t$, where $\mathbf{s}_t \in \mathbb{R}^{25}$, contains the same information as that in Pick-And-Place, but the object's information, except its relative position, is set to 0.

An action $\mathbf{a}_t$ in all environments, where $\mathbf{a}_t \in \mathbb{R}^4$, contains the position variation of the end-effector and the distance between the two grippers.

#### A.2.2 PREDEFINED KNOWLEDGE

KG1 and KG2 for OpenAI robotic environments are hand-crafted rules that output a mean and standard deviation of an action given a state. We set $\varepsilon$ to be 0.03 for Pick-And-Place and the log standard deviation to be -1 for both environments. We provide our code of defining the rules here. Note that we will apply the softmax function to an output to make it a probability vector.

```python
if env_type == 'reach':
    kg_num = 1

    kg_mean_actions = th.zeros([batch_size, kg_num, action_dim])
    #Reach, KG1, move straightly to the goal
    kg_mean_actions[:,0,:3] = desired_goal - current_grip_pos

elif env_type == 'pick_and_place':
    kg_num = 2

    kg_mean_actions = th.zeros([batch_size, kg_num, action_dim])

    rel_pos, g_state = obs['observation'][:,6:9], obs['observation'][:,9].unsqueeze(1)
    rel_pos_norm = th.linalg.norm(rel_pos, dim=-1, keepdim=True) #batch size x 1

    #Pick-And-Place, KG2, move straightly to the object
    kg_mean_actions[:,0,:3] = rel_pos * (rel_pos_norm >= 0.03).int()
    kg_mean_actions[:,0,-1] = (1. * (rel_pos_norm >= 0.03).int() + g_state * (rel_pos_norm < 0.03).int())[:,0]

    #Pick-And-Place, KG1, move straightly to the goal
    kg_mean_actions[:,1,:3] = (desired_goal - current_grip_pos) * (rel_pos_norm < 0.03).int()
    kg_mean_actions[:,1,-1] = (-1. * (rel_pos_norm < 0.03).int() + g_state * (rel_pos_norm >= 0.03).int())
      [:,0]

kg_log_std = th.ones([batch_size, kg_num, action_dim]) * -1
```

#### A.2.3 BASELINE ALGORITHMS

**BC.** Since KG1 and KG2 are not in conflict with each another, we can combine these two knowledge mappings (KG1+KG2). The results of BC is obtained by applying KG1+KG2 in both environments.

**SAC-BC.** The external knowledge policy KG1+KG2 can generate demonstrations to guide an agent. We use this policy to collect demonstration data with $2 \times 10^5$ steps for Reach and $10^6$ steps for Pick-And-Place. Then we follow the method proposed in Nair et al. (2018) to train an agent. The demonstration batch size is set to be 128 for both environments. The coefficient of the BC loss is set to be $10^{-3}$ for both environments. Note that we only apply this loss to the mean of the policy's output, and the standard deviation is not learned with any supervised signal.

#### A.2.4 MODEL ARCHITECTURE

Each external knowledge embedding is generated by the PyTorch module, `nn.Embedding`, with $d_k = 4$ for both Reach and Pick-And-Place. The inner actor network, $\phi_{in}$, is an MLP for both tasks. $\phi_{in}$ in Reach is with two hidden layers and a hidden size of 64 units, and that in Pick-And-Place is with three hidden layers and a hidden size of 512 units. The critic network, $\phi_c$, is with the same architecture as that of $\phi_{in}$. The inner key network, $\phi_{\mathbf{k}}$, is an MLP for both tasks. Different from the inner key network in MiniGrid environments, $\phi_{\mathbf{k}}$ in

OpenAI robotic environments does not share layers with $\phi_{in}$. $\phi_{\mathbf{k}}$ in Reach is with one hidden layer and a hidden size of 32 units, and that in Pick-And-Place is with two hidden layers and a hidden size of 64 units. The query network, $\phi_{\mathbf{u}}$, is with the same architecture as that of $\phi_{\mathbf{k}}$. All hidden layers in $\phi_{in}, \phi_c, \phi_{\mathbf{k}}$, and $\phi_{\mathbf{u}}$ are followed by ReLU activation.

### A.2.5 HYPERPARAMETERS

Each training trial in Section 5.2 is run with 5 different random seeds, and each testing trial is run with 100 different random initializations. For the underlying SAC algorithm in Section 5.2, we follow the implementation of Stable-Baselines3 (SB3) Raffin et al. (2021). The training timesteps for Reach and Pick-And-Place are 20K and 1M respectively. The learning rates are $10^{-3}$ and $5 \times 10^{-4}$ respectively. The batch sizes are 256 and 2048 respectively. The sizes of the replay buffer are both 1M. The discount factors $\gamma$ are both 0.95. The temperature of entropy is adjusted automatically in both environments as described in Haarnoja et al. (2018).

