# OpenReview forum: "Knowledge-Grounded Reinforcement Learning"
_ICLR.cc/2023/Conference — Submitted to ICLR 2023_

### Official Review · Reviewer_ce5j · 2022-10-22

**Confidence:** 5
**Correctness:** 1
**Technical Novelty And Significance:** 1
**Empirical Novelty And Significance:** 1
**Recommendation:** 3

**Clarity, Quality, Novelty And Reproducibility:**

The paper does not present a novel idea. The experiments do not reflect the broad range of problems claimed to be solved by the authors. The authors do not consider alternative architectures and justify why the proposed design was chosen.

**Strength And Weaknesses:**

Strengths:
* Incorporating external knowledge into RL policy is an important problem.
* The method outperforms proposed baselines.
* The paper is easy to read.

Weaknesses:
* The experimental settings and results do not support claims made by the authors. First, the author "knowledge" as "*can be any commonsense, suggestion, guideline, principle, specification, rule, restriction, or previously learned knowledge described in formal or informal language*", but the knowledge in the experiments is simply commands or high-level options. The authors only construct instruction-following tasks rather than knowledge-incorporating tasks, which are much broader in scope and more difficult to solve.
* The proposed method is not novel. Attentional architecture is the de-facto choice for building instruction-following agents [1]. The authors simply train the agent end-to-end without proposing any specific techniques that help the policy incorporate the knowledge effectively. Knowledge such as constraints or environment specifications may require more special treatments than simple attention mechanisms [2,3].
* There are other ways to design the attention mechanism but the authors do not explain why they choose the presented one. They also do not compare with alternative designs in the experiments. For example, instead of mixing the policies, I can mix at internal representation level h = sum(w_i * h_i), where each h_i is the representation of a piece of knowledge. Then I construct a single policy based on the mixed representation. This avoids having to perform the gumbel-softmax trick.

[1] https://arxiv.org/pdf/1711.07280.pdf
[2] https://arxiv.org/pdf/2010.05150.pdf
[3] https://arxiv.org/pdf/2101.07393.pdf





**Summary Of The Paper:**

The paper proposes an architecture that can leverage external knowledge to make better decisions in RL problems. The architecture implements an attention mechanism that dynamically chooses to attend to an internal knowledge mapping or external ones. It is trained end-to-end via the Gumbel-softmax trick. Experiments are conducted on grid-world and robotic tasks.

**Summary Of The Review:**

The merits of the paper are marginal. I do think the idea presented would be significant for the community. I am learning towards rejecting the paper.

=====After Rebutal=====

I have read the authors' response and decided to keep my current score. My main point of rejection is that the paper over-claims the generality of their methods while not having experiments to back up and avoiding comparing with related work. The definition of knowledge, which can be "commonsense, suggestion, guideline, principle, specification, rule, or restriction, described as a knowledge mapping" is very vague without defining what the mapping actual is (e.g. how is that policy related to the optimal policy?). I suggest the authors revise the scope of their claims and provide a more specific specification of their setting.

---

> ### Author Response · Authors · 2022-11-17
> **Response to Reviewer ce5j**
>
> 1. **The experimental settings and results do not support claims made by the authors. First, the author "knowledge" as "can be any commonsense, suggestion, guideline, principle, specification, rule, restriction, or previously learned knowledge described in formal or informal language", but the knowledge in the experiments is simply commands or high-level options. The authors only construct instruction-following tasks rather than knowledge-incorporating tasks, which are much broader in scope and more difficult to solve.**
>
> Thank you for telling us your concern. We would like to refer you to the general response above, “Response to Shared Comments (1/2) - 2,” and Section 3 of our updated manuscript, where we define knowledge to be state-action mappings instead of text.
>
> Also, this work is not about instruction following but about guiding an agent with an arbitrary set of external knowledge policies.
>
> 2. **The proposed method is not novel. Attentional architecture is the de-facto choice for building instruction-following agents [1]. The authors simply train the agent end-to-end without proposing any specific techniques that help the policy incorporate the knowledge effectively. Knowledge such as constraints or environment specifications may require more special treatments than simple attention mechanisms [2,3].**
>
> Thank you for telling us your concern. We would like to refer you to the general response above, “Response to Shared Comments (1/2) - 3”.
>
> This work is not about text-based RL or learning an instruction-following agent. Also, our method is novel since, to the best of our knowledge, no prior work has considered incorporating external policies in this way.
>
> 3. **There are other ways to design the attention mechanism but the authors do not explain why they choose the presented one. They also do not compare with alternative designs in the experiments. For example, instead of mixing the policies, I can mix at internal representation level h = sum(w_i * h_i), where each h_i is the representation of a piece of knowledge. Then I construct a single policy based on the mixed representation. This avoids having to perform the gumbel-softmax trick.**
>
> Thank you for telling us your concern. We would like to refer you to Section 4 of our updated manuscript, where we explain the idea of designing the proposed method. The current design allows us to disentangle the knowledge-policy representation and selection mechanism for knowledge recomposition/rearrangement, which is not achievable by mixing knowledge at the representational level.

---

### Official Review · Reviewer_uBBQ · 2022-10-24

**Confidence:** 4
**Correctness:** 2
**Technical Novelty And Significance:** 2
**Empirical Novelty And Significance:** 2
**Recommendation:** 3

**Clarity, Quality, Novelty And Reproducibility:**

- Clarity: As stated in the ‘strengths’ section, the proposed method is explained well, and the relevant literature section makes clear where the paper is placed relative to the rest of the field. The title (which is used throughout the paper) is a bit misleading, in the sense that the ‘knowledge’ used to ‘ground’ the RL in fact consists of pre-baked policies. While it is technically possible to argue that this is a form of grounding RL in knowledge, it would be more accurate to phrase it in terms of something along the lines of ‘sub-policy re-use’.
- Quality: As stated in the weaknesses, there are major gaps in both the usefulness of the method that is demonstrated, and the scope of the experimental evaluation. The idea of reusing policies has merit in principle, but it is also difficult to execute well, and this paper does not reach the bar of a convincing demonstration. To start with, the authors don’t address the question of how to pick pre-trained policies for a particular task. They don't show how to automate a system based on re-using policies, how to create diversity in the learned policies, or how to deal with the probably very large dictionary of policies that agents would have to learn to navigate. As it is, the paper does nothing to convince the reader that the proposed method would go beyond toy tasks where it is easy to provide pre-baked policies.
- Novelty: while the HRL and options literature is referenced, there is a degree of disconnect between the claims about the proposed method made in the section discussing HRL/options, and the experiments and implementation that is actually presented. Paraphrasing, the claim is made that re-used policies in KGRL can be much more varied than skills in HRL or options. That might in theory be true, but in practice the pre-baked policies provided to the agents in the paper are very good examples of skills or options.
- Reproducibility: no concerns.

**Strength And Weaknesses:**

- Weakness: experimental evaluation. The MiniGrid environment is clearly a toy one, which might be suited for a proof of concept, but is not sufficient for a serious evaluation. The two robotics tasks are quite elementary as well; more complex tasks exist, and would be very natural applications for the proposed method - a very simple example being stacking blocks.
- Weakness: the method as it is used in the paper relies heavily on hand-crafted pre-baked policies. The paper does not show how agents learn policies that can be transferred to agents learning other tasks.
- Strength: the related literature section is fairly comprehensive and covers a good number of related topics.
- Strength: the proposed method is explained clearly.

**Summary Of The Paper:**

The paper introduces an agent architecture intended to facilitate the reuse of existing knowledge, called ‘knowledge grounded RL’. The knowledge grounding consists of a dictionary with randomly initialized learnable keys paired with hard-coded policies (these could possibly also be pre-trained, in principle) which is available to the agent. There is one additional policy in the dictionary which is not hard-coded or pre-trained, but trained online. Action selection happens by attending over this dictionary.

The proposed architecture is evaluated on six MiniGrid and two robotics tasks, with a specific set of hard-coded policies for each environment. With the right set of pre-baked policies, the proposed architecture outperforms the chosen baselines on both environments.


**Summary Of The Review:**

The idea of re-using existing knowledge, or more specifically, of policies trained or defined previous to the agent's initialization, has merit. However, the implementation and evaluation of that idea in this paper falls well short of the standard for novelty and quality for an academic contribution. The method relies too much on hand-crafted solutions, and the evaluation tasks are too simple and too few and unvaried. I recommend rejection.

---

> ### Author Response · Authors · 2022-11-17
> **Response to Reviewer uBBQ**
>
> 1. **Weakness: experimental evaluation. The MiniGrid environment is clearly a toy one, which might be suited for a proof of concept, but is not sufficient for a serious evaluation. The two robotics tasks are quite elementary as well; more complex tasks exist, and would be very natural applications for the proposed method - a very simple example being stacking blocks.**
>
> Thank you for telling us your concern. We would like to refer you to the general response above, “Response to Shared Comments (1/2) - 2”. The environments we chose are not all toy environments, and their different levels of difficulty allow us to demonstrate the important properties of our KGRL method.
>
> 2. **Weakness: the method as it is used in the paper relies heavily on hand-crafted pre-baked policies. The paper does not show how agents learn policies that can be transferred to agents learning other tasks.**
>
> Thank you for telling us your concern. We would like to refer you to Section 5.1 “Analysis of Reusing, Recomposing, and Rearranging Knowledge” of our manuscript, where we have demonstrated that our method can simultaneously incorporate different forms of external knowledge policies (hand-crafted rules and previously-learned neural networks).
>
> 3. **The title (which is used throughout the paper) is a bit misleading, in the sense that the ‘knowledge’ used to ‘ground’ the RL in fact consists of pre-baked policies. While it is technically possible to argue that this is a form of grounding RL in knowledge, it would be more accurate to phrase it in terms of something along the lines of ‘sub-policy re-use’.**
>
> Thank you for the suggestion. We would like to refer you to the general response above, “Response to Shared Comments (1/2) - 1,” about the differences between KGRL and policy reuse/transfer.
>
> 4. **Quality: As stated in the weaknesses, there are major gaps in both the usefulness of the method that is demonstrated, and the scope of the experimental evaluation. The idea of reusing policies has merit in principle, but it is also difficult to execute well, and this paper does not reach the bar of a convincing demonstration.**
>
>     **To start with, (1) the authors don’t address the question of how to pick pre-trained policies for a particular task. (2) They don't show how to automate a system based on re-using policies, (3) how to create diversity in the learned policies, or (4) how to deal with the probably very large dictionary of policies that agents would have to learn to navigate.**
>
>     **As it is, the paper does nothing to convince the reader that the proposed method would go beyond toy tasks where it is easy to provide pre-baked policies.**
>
> Thank you for telling us your concern. We would like to refer you to the general response above, “Response to Shared Comments (1/2) - 2,” and Section 5 of our manuscript, where we have done multiple analyses to discuss the properties of our proposed method.
>
> We answer the individual questions raised in this comment as follows:
>
> (1) As stated in Section 3 of our manuscript, any state-action mapping that may guide an agent to complete a task can be external knowledge. In our experiments, we chose the most naive strategies as external knowledge to demonstrate that picking them requires little effort.
>
> (2) We have shown this in Section 5.1 “Analysis of Reusing, Recomposing, and Rearranging Knowledge” of our manuscript.
>
> (3) We are not sure if we understand this correctly, but the goal of KGRL is to incorporate an arbitrary set of external knowledge policies into RL instead of creating the diversity of a learned policy.
>
> (4) We demonstrate in Figure 3 of our updated manuscript (or Figure 2 of our original one) that our KGRL agent can navigate through an external knowledge set with redundant (not task-related) knowledge. As stated in Section 6, we leave building a large knowledge set as future work.
>
> 5. **Novelty: while the HRL and options literature is referenced, there is a degree of disconnect between the claims about the proposed method made in the section discussing HRL/options, and the experiments and implementation that is actually presented. Paraphrasing, the claim is made that re-used policies in KGRL can be much more varied than skills in HRL or options. That might in theory be true, but in practice the pre-baked policies provided to the agents in the paper are very good examples of skills or options.**
>
> Thank you for telling us your concern. We would like to refer you to the general response above, “Response to Shared Comments (1/2) - 2,” and Figure 3 of our updated manuscript (or Figure 2 of our original one), where we have demonstrated that our KGRL method can handle redundant (not task-related) knowledge.

---

> > ### Comment · Reviewer_uBBQ · 2022-12-09
> > **Thank you for your response**
> >
> > Thank you for your response, and apologies for my late reaction!
> >
> > - Regarding task difficulty: I acknowledge your statement that the tasks are not all toy. My point was that your method is aimed at complex tasks, where reusing existing knowledge is important, but the tasks you use are not all that complex. E.g. https://openreview.net/pdf?id=U0Q8CrtBJxJ, or some tasks from https://github.com/openai/retro, offer significantly more complex tests, where the usefulness of the proposed method could be demonstrated much more convincingly.
> > - Section 5.1 is a demonstration of transferring trained policies, thank you (apologies if I missed that the first time!). I would not necessarily say that hand-crafted rules and previously-learned networks are different forms of knowledge, since they are both "policy-shaped" knowledge, but transfer it is.
> > - Regarding the title: "knowledge" covers more than policies/options, although those are forms of knowledge. By using the word knowledge, you imply that you do more than reusing policies.
> > - On point 4:
> >   1. Yes, any policy can be knowledge, but you pick some specific ones - you say yourself that you 'designed' them, even if you designed them to be suboptimal, and that makes your method dependent on their availability, which in turn makes it important to specify how that availability can be guaranteed for more complex tasks.
> >   2. Section 5.1 does not show any automation, it does show reuse. With your demonstrations, there is still a human needed to start every iteration, which is a limitation of your method.
> >   3. If the agent is going to reuse policies for goals that are presumably unknown at the time of learning those reusable policies, it would be good if they had some diversity, given that the goal they will be reused for is not known. If beforehand the designer has to know which sub-policies are sufficient for an efficient solution, that is another limitation of your method.
> >   4. Your sets of policies are very small, which is possible because your tasks are relatively simple. If you would apply your method to more complex problems, with more subgoals, and potentially several iterations of learning and reusing, the set of policies is likely to become much bigger, which would mean the KGRL agent has to solve a hard exploration problem. Leaving this as future work is a choice of yours, of course, but making plausbile that it can be done in the future is important in that case.
> > - Regarding the policies/options versus KGRL claim: the ability to learn to ignore redundant knowledge is a good thing, but does not support your claim that your method is more general than HRL.
> >
> > All in all, my assessment that while the idea behind the paper has merit, the demonstration is not convincing enough to clear the bar, remains.

---

> > > ### Author Response · Authors · 2022-12-11
> > > **Thank you for the response!**
> > >
> > > Thank you for the comments and suggestions. We would like to refer you to our responses below:
> > >
> > > 1. **Regarding task difficulty: I acknowledge your statement that the tasks are not all toy. My point was that your method is aimed at complex tasks, where reusing existing knowledge is important, but the tasks you use are not all that complex. E.g. https://openreview.net/pdf?id=U0Q8CrtBJxJ, or some tasks from https://github.com/openai/retro, offer significantly more complex tests, where the usefulness of the proposed method could be demonstrated much more convincingly.**
> > >
> > > Thank you for the suggestion. We will consider adding more experiments in our revised manuscript.
> > >
> > > 2. **Section 5.1 is a demonstration of transferring trained policies, thank you (apologies if I missed that the first time!). I would not necessarily say that hand-crafted rules and previously-learned networks are different forms of knowledge, since they are both "policy-shaped" knowledge, but transfer it is.**
> > >
> > >     **Regarding the title: "knowledge" covers more than policies/options, although those are forms of knowledge. By using the word knowledge, you imply that you do more than reusing policies.**
> > >
> > > We understand that the term “knowledge” can mean something more than “a strategy that maps from states to actions” and will consider revising our manuscript to better clarify the terminology. Yet, we would like to point out that, as stated in the general response, “Response to Shared Comments (1/2) - 1”, our definition of knowledge is more explicitly given and follows a series of previous works.
> > >
> > > In addition, “different forms of knowledge” in our manuscript and responses mean that their representations are different, e.g., they can be hand-crafted rules, neural networks, or fuzzy logic.
> > >
> > > 3. **Yes, any policy can be knowledge, but you pick some specific ones - you say yourself that you 'designed' them, even if you designed them to be suboptimal, and that makes your method dependent on their availability, which in turn makes it important to specify how that availability can be guaranteed for more complex tasks.**
> > >
> > > Thank you for telling us your concern. However, we would like to point out that the availability of external knowledge is beyond the scope of this work. In Section 1, we mention that KGRL aims to answer the question of how an agent follows an arbitrary set of external knowledge instead of how external knowledge is made available to an agent. In Section 3, the problem formulation of KGRL, we assume that a set of external knowledge is given to the agent.
> > >
> > > 4. **Section 5.1 does not show any automation, it does show reuse. With your demonstrations, there is still a human needed to start every iteration, which is a limitation of your method.**
> > >
> > > Thank you for telling us your concern. However, we would like to indicate that automating multiple iterations of KGRL is beyond the scope of this work. We assume that a set of external knowledge is given to an agent at each iteration. Even with this assumption, the problem of KGRL is challenging enough as explained in Sections 1 and 2.
> > >
> > > 5. **Your sets of policies are very small, which is possible because your tasks are relatively simple. If you would apply your method to more complex problems, with more subgoals, and potentially several iterations of learning and reusing, the set of policies is likely to become much bigger, which would mean the KGRL agent has to solve a hard exploration problem. Leaving this as future work is a choice of yours, of course, but making plausbile that it can be done in the future is important in that case.**
> > >
> > > Thank you for telling us your concern. Although we leave incorporating a very large policy set as future work, our method is not limited to a small set of policies. As each external policy is represented by a learned embedding, and the attention mechanism is not affected by the size of the policy set, theoretically a very large number of policies can be simultaneously incorporated.
> > >
> > > 6. **Regarding the policies/options versus KGRL claim: the ability to learn to ignore redundant knowledge is a good thing, but does not support your claim that your method is more general than HRL.**
> > >
> > > Thank you for telling us your concern. We do not claim that our method is more general than HRL. The goals of KGRL and HRL are different. KGRL focuses on referring to external guidance in arbitrary forms and extracting a reusable knowledge embedding and query. On the other hand, HRL focuses on decomposing a long-horizon task into a hierarchy of subtasks and learning a high-level policy that chooses optimal subtasks as high-level actions. In Section 2, we compare the differences between KGRL and HRL only from the policy-reuse perspective.

---

### Official Review · Reviewer_4vDB · 2022-10-30

**Confidence:** 2
**Correctness:** 3
**Technical Novelty And Significance:** 2
**Empirical Novelty And Significance:** 2
**Recommendation:** 5

**Clarity, Quality, Novelty And Reproducibility:**

The work is described clearly and is of good quality. The novelty is lacking as several literature work on reinforcement learning with external knowledge is missing (see comments above in Strengths and Weaknesses). Not enough details is provided to reproduce the experiments. For example, it's not clear how the external knowledge is mapped into executable actions. It's also not clear how to integrate other forms of knowledge in general. E.g., domain rules that could be represented in formal logic or even text.

**Strength And Weaknesses:**

Strengths

The proposed framework demonstrates superior performance in terms sample efficiency on tasks used and does so with different policy training algorithms.

Weaknesses

The paper misses related work such as Kimura, Daiki, et al. "Reinforcement learning with external knowledge by using logical neural networks." arXiv preprint arXiv:2103.02363 (2021) and Murugesan, Keerthiram, et al. "Text-based RL Agents with Commonsense Knowledge: New Challenges, Environments and Baselines." AAAI. 2021.

Text-based RL benchmarks such as those used in the above works (e.g., Textworld, Zork, etc) are good candidates for this work. The proposed work would be much strengthened it also used these benchmarks and compared to SOTA methods there.

Although the proposed framework is described generally, the external knowledge used in the experiments is task specific and could be learned and re-used by existing transfer techniques. Why shouldn't curriculum learning be equally successful in this setting? The experiments seem to be demonstrating a way of recombining/re-using previously learned knowledge, which could be learned internally only using e.g., some curriculum.

The implementation details are very general. I don't think it would be easy to reproduce the results. For example, details are provided (Appendix) about how to build embeddings for external knowledge but it's not clear how that external knowledge is encoded. This is supposed to be done by the external knowledge mappings but not details are provided for any of the experiments.

Lastly, it's not clear what the internal policy is learning. It seems to be that the internal policy is learning to explore the environment and relies on external knowledge for task execution. So, in my opinion, the internal policy is not learning the policy to complete the tasks but the whole model is learning how to use external knowledge without building any internal knowledge for reuse later.

Also, I'm not sure the learned policy is interpretable. As shown in the experiments, we can see when the agent chooses to use which piece of knowledge but the individual pieces of knowledge are not interpretable in and of themselves, which is generally the issue of interpretability.

**Summary Of The Paper:**

The paper proposes a framework for incorporating background knowledge into reinforcement learning, Knowledge-Grounded RL. The approach uses an embedding-based attention-mechanism model that learns to attend to external and internal knowledge based on observations. The proposed framework is agnostic to the policy training algorithm, and was demonstrated using PPO and SAC algorithms.

The proposed framework is evaluated on several tasks from two benchmarks: navigation tasks from the MiniGrid environment and pick-and-place manipulation tasks from OpenAI-Robotics benchmark. In both sets of tasks, the proposed framework superior or competitive performance in terms of sample efficiency.

**Summary Of The Review:**

The paper address an important aspect of AI: how to incorporate external knowledge into a learning agent. It proposes an embedding-based actor model that learns to attend to different sources of knowledge and evaluates this on several benchmarks. However, it misses important pieces of the literature on the same topic and doesn't evaluate against SOTA in the space of reinforcement learning with external knowledge. It also misses important benchmarks in the area.

---

> ### Author Response · Authors · 2022-11-17
> **Response to Reviewer 4vDB**
>
> 1. **The paper misses related work such as Kimura, Daiki, et al. "Reinforcement learning with external knowledge by using logical neural networks." arXiv preprint arXiv:2103.02363 (2021) and Murugesan, Keerthiram, et al. "Text-based RL Agents with Commonsense Knowledge: New Challenges, Environments and Baselines." AAAI. 2021.**
>
>     **Text-based RL benchmarks such as those used in the above works (e.g., Textworld, Zork, etc) are good candidates for this work. The proposed work would be much strengthened it also used these benchmarks and compared to SOTA methods there.**
>
> Thank you for the suggestion. We would like to refer you to the general response above, “Response to Shared Comments (2/2) - 6,” and Section 3 of our updated manuscript, where we discuss the differences between text-based RL and KGRL and formally define KGRL respectively. In short, text-based RL is completely different from KGRL in terms of the goals and the definition of knowledge.
>
> 2. **Although the proposed framework is described generally, the external knowledge used in the experiments is task specific and could be learned and re-used by existing transfer techniques. Why shouldn't curriculum learning be equally successful in this setting? The experiments seem to be demonstrating a way of recombining/re-using previously learned knowledge, which could be learned internally only using e.g., some curriculum.**
>
> Thank you for the question. The goal of KGRL is to incorporate an arbitrary set of existing policies in multiple forms. Nowadays, there are lots of ready-to-use policies, either trained by others or easy to define. From a resource-saving perspective, it is more efficient to use them directly instead of learning them from scratch.
>
> Also, we would like to point out that the external knowledge we use in this work is not task-specific when viewing it from a higher level, e.g., “go straightly to the goal” or “pick up an object”, which can be shared by many tasks.
>
> 3. **The implementation details are very general. I don't think it would be easy to reproduce the results. For example, details are provided (Appendix) about how to build embeddings for external knowledge but it's not clear how that external knowledge is encoded. This is supposed to be done by the external knowledge mappings but not details are provided for any of the experiments.**
>
> Thank you for indicating the clarity issue. We would like to refer you to the general response above, “Response to Shared Comments (2/2) - 4,” and Appendix A of our updated manuscript, where we add more experimental details, including the code for defining the external knowledge.
>
> 4. **Lastly, it's not clear what the internal policy is learning. It seems to be that the internal policy is learning to explore the environment and relies on external knowledge for task execution. So, in my opinion, the internal policy is not learning the policy to complete the tasks but the whole model is learning how to use external knowledge without building any internal knowledge for reuse later.**
>
> Thank you for indicating the clarity issue. We would like to refer you to the general response above, “Response to Shared Comments (2/2) - 5,” where we explain what the inner policy learns. In short, the inner policy learns its own strategy to partially or fully complete a task, and we provide several experimental results in the manuscript.
>
> 5. **Also, I'm not sure the learned policy is interpretable. As shown in the experiments, we can see when the agent chooses to use which piece of knowledge but the individual pieces of knowledge are not interpretable in and of themselves, which is generally the issue of interpretability.**
>
> Thank you for telling us your concern. Since we encode each knowledge policy into an embedding space, they can be interpretable in that space. For example, if two knowledge policies adopt similar strategies, they will be closer to each other in the embedding space. A similar idea of interpretable skills in an embedding space can be found in [1].
>
> 6. **Not enough details is provided to reproduce the experiments. For example, it's not clear how the external knowledge is mapped into executable actions. It's also not clear how to integrate other forms of knowledge in general. E.g., domain rules that could be represented in formal logic or even text.**
>
> Thank you for indicating the clarity issue. We would like to refer you to Appendix A of our updated manuscript, where we add more experimental details. External knowledge in different forms, e.g., fuzzy logic, can be directly incorporated into our KGRL method as long as they are state-action mappings. This is because we learn a uniform representation for each external knowledge policy.
>
> [References]
>
> [1] Florensa, Carlos, Yan Duan, and Pieter Abbeel. "Stochastic neural networks for hierarchical reinforcement learning." arXiv preprint arXiv:1704.03012 (2017).

---

### Official Review · Reviewer_quir · 2022-10-31

**Confidence:** 4
**Correctness:** 3
**Technical Novelty And Significance:** 3
**Empirical Novelty And Significance:** 3
**Recommendation:** 5

**Clarity, Quality, Novelty And Reproducibility:**

While the main idea of this work is straightforward and the English writing is OK, many important details of the proposed method are missing, making many parts of this work difficult to follow.

**Strength And Weaknesses:**

Strength:
1. The idea that agents learn to follow external guidelines and develop their own policies makes sense.
2. The related work description is very detailed.

Weaknesses:
1. As mentioned in Section 4.1, "We learn a key for each external knowledge mapping...", how to learn these keys? I did not find any descriptions.
2. In Figure 1, $k_in$ is generated by the Inner Actor, which can be learned during the interaction. However, there is no input about $k_gi$. Is it missing? Or please give some more description about my misunderstanding.
3. In the paper, PPO and SAC are used as the baseline algorithm respectively, and I hope that more baseline algorithms can be combined to illustrate the generality.
4. At the beginning of Section 5, this paper claims to answer the question "(1) Can our KGRL agent learn effective new knowledge by referring to an arbitrary set of external guidance?". Does the term 'new knowledge' simply refer to the stringing together of old knowledge? If so, the use of the word "new" is not appropriate, and if not, could you please give some experimental examples?
5. Since Atari games often appear in reinforcement learning algorithm articles as a basic verification environment, it is hoped that this paper can also do a brief algorithm verification in this type of environment.

**Summary Of The Paper:**

This paper defines a knowledge-ground reinforcement learning (KGRL) problem that an agent learns to follow external guidelines and develop its own policy. It also provides a realization of this problem by using an attention-based actor model which can switch between either a learnable internal policy or external knowledge. Experiments on MiniGrid and OpenAI-Robotics show the KGRL agent's effectiveness.

**Summary Of The Review:**

I like the main idea behind this work, and the proposed actor model architecture looks reasonable. However, since many important details of the proposed method are missing, and there is no source code provided to check how the proposed model is implemented, I thus can not recommend acceptance of this work in its current form. I strongly encourage the authors to re-organize the manuscript's content and re-write the whole paper for better understanding by the audiences.

---

> ### Author Response · Authors · 2022-11-17
> **Response to Reviewer quir**
>
> 1. **As mentioned in Section 4.1, "We learn a key for each external knowledge mapping...", how to learn these keys? I did not find any descriptions.**
>
> Thank you for indicating the clarity issue. We would like to refer you to Section 4 of our updated manuscript, where we add more explanation about the proposed method. In short, all learnable components (the keys, query, and inner policy) are end-to-end learned by an RL algorithm.
>
> 2. **In Figure 1, k_{in} is generated by the Inner Actor, which can be learned during the interaction. However, there is no input about k_{g_i}. Is it missing? Or please give some more description about my misunderstanding.**
>
> Thank you for indicating the clarity issue. We would like to refer you to Section 4 of our updated manuscript, where we explain why an external knowledge key is state-independent. In short, since it represents an entire external knowledge policy, and the strategy of this policy is fixed, it should not depend on states or actions.
>
> 3. **In the paper, PPO and SAC are used as the baseline algorithm respectively, and I hope that more baseline algorithms can be combined to illustrate the generality.**
>
> Thank you for the suggestion. We would like to refer you to Section 5 of our updated manuscript, where we add two more baselines, BC and RL+BC, to compare with our KGRL method.
>
> 4. **At the beginning of Section 5, this paper claims to answer the question "(1) Can our KGRL agent learn effective new knowledge by referring to an arbitrary set of external guidance?". Does the term 'new knowledge' simply refer to the stringing together of old knowledge? If so, the use of the word "new" is not appropriate, and if not, could you please give some experimental examples?**
>
> Thank you for the question. We would like to refer you to the general response above, “Response to Shared Comments (2/2) - 5,” for the answer to this question.
>
> 5. **Since Atari games often appear in reinforcement learning algorithm articles as a basic verification environment, it is hoped that this paper can also do a brief algorithm verification in this type of environment.**
>
> Thank you for the suggestion. We chose the MiniGrid and OpenAI robotic environments since they provide a series of tasks that allow us to demonstrate the knowledge recomposition/rearrangement/reuse property of our method.
>
> 6. **While the main idea of this work is straightforward and the English writing is OK, many important details of the proposed method are missing, making many parts of this work difficult to follow.**
>
> Thank you for indicating the clarity issue. We would like to refer you to the general response above, “Response to Shared Comments (2/2) - 4,” and Section 4 and Appendix A of our updated manuscript, where we add more details about the proposed method and experiments.

---

### Official Review · Reviewer_c6pa · 2022-11-02

**Confidence:** 4
**Correctness:** 2
**Technical Novelty And Significance:** 2
**Empirical Novelty And Significance:** Not applicable
**Recommendation:** 3

**Clarity, Quality, Novelty And Reproducibility:**

The writing and presentation of the proposed method is almost clear. The central notion (i.e., knowledge) is kind of misleading and improper. The proposed method is lack of novelty. A few key claims and statements are not well supported or explained.

For reproductibility, some details on implementation and experiments are provided in the appendix. Some key details of the pre-defined external knowledge policies are missing (e.g., $\epsilon$ in Section 5.2). The source codes are not provided.


**Strength And Weaknesses:**

$\textbf{Strengths:}$
+ The writing and presentation of the proposed method is almost clear.
+ The related work includes multiple related domains.
+ The experiments are conducted from multiple aspects.

&nbsp;

$\textbf{Weaknesses:}$
- The major issue is on the central notion of this paper, i.e., Knowledge or KGRL. I think it is improper and kind of misleading to use the notion of knowledge through the paper. After reading the paper, I recommend the authors to use the notion e.g., Policy Reuse/Transfer, which is more accurate.
  - The authors emphasize many times in this paper like the claim ‘all knowledge can be arbitrarily recomposed, rearranged, and reused anytime in the learning and inference stages’. However, the concrete form of external knowledge considered is the external knowledge policy.
  - In other words, acctually the main content of this work is on how to incorporate, reuse and transfer existing policies in RL. For KGRL at least in my opinion, the key part is how different (or several common) forms of external knowledge are represented, after which how to use the knowledge is to be considered. However, this part is not included in this work and in the experiments, the external knowledge is implemented by the pre-defined policies.
  - Given the external knowledge policies, the proposed method in Section 4 is not novel to me. The self-attention mechanism and the end-to-end update are straightforward.
- The experiments provide few impressive results, especially when the pre-defined external knowledge policies are strong and the environments are not challenging.
- Important related work is missing. For example, Knowledge Guided Policy Network (KoGuN) [1] propose a similar idea, where suboptimal external knowledge is represented by trainable module based on fuzzy logic and then is incorporated into PPO policy network serving as a refine module.

&nbsp;

Reference:

[1] Peng Zhang, Jianye Hao, Weixun Wang, Hongyao Tang, Yi Ma, Yihai Duan, Yan Zheng:
KoGuN. Accelerating Deep Reinforcement Learning via Integrating Human Suboptimal Knowledge. IJCAI 2020



&nbsp;

$\textbf{Questions:}$

1) The authors claim “current learning-from-demonstration and knowledge-reuse approaches lack the flexibility to rearrange and recompose different demonstrations or knowledge, so they cannot dynamically adapt to environmental changes”. Do the author mean the ability of zero-shot adaptation or generalization? To my knowledge, policy reuse and policy transfer are well-studied domain in this direction. Can the authors justify this claim?
2) The authors claim “These two research directions do not enable incorporating external guidance from different sources into learning, so their transferred knowledge is limited to the one learned among similar environments” and “KGRL allows an agent to follow and reuse an arbitrary set of knowledge obtained from different sources in a new task, and these sources can be very different from the new task”. What do the ‘different sources’ mean? External knowledge in different forms? It is confusing to me since this paper does not address how arbitrary forms of external knowledge are transformed exactly into the external knowledge policies. Instead, the external knowledge policies are given in the setting considered in this work.
3) Using a state-dependent key for the inner policy makes sense to me. However, I recommend the authors to add a KGRL variant without state-dependent key as an additional baseline in the experiments for more complete evaluation and abalation.
4) I also have concern on the claim “we disentangle the decision component (the query) and the knowledge representations (the keys)”. Indeed, they are separated but they are updated in an entangled form regarding the policy gradients.
5) I think there is a straightforward alternative to the gumble-softmax way to acting with the inner policy and multiple external policies. Concretely, to sample an action for each policy (with reparameterization) and then to mix them according to a softmax distribution of $w$. Why is this way not considered? Or what is the drawback of this alternative way?
6) Are the external knowledge policies used in the experimenst stochastic or deterministic?
7) It is strange to see the very small error bars in Figure 2a and 2b (in fact, I cannot tell if there is a error bar in Figure 2b).
8) In Figure 2e, why do PPO-KGRL (KG3) and PPO-KGRL (KG1+KG2) perform worse than PPO while PPO-KGRL (all) outperforms PPO?
9) In Section 5.2, it seems that KG1 is already optimal in Reach task and KG1 + KG2 is optimal in Pick-and-Place. Can the authors provide more explanation on this point? Similarly, KG1 + KG2 + KG3 is also optimal in 5x5 environments since the view range is also 5x5.
10) The authors mention “indicating that a KGRL agent learns to not only follow the guidance but also mimic its strategy. This imitation process allows an agent to develop an inner policy that outperforms external strategies”. Can the authors provide more explanation on how such imitation happen under KGRL and why imitation can be better than the external policies?


**Summary Of The Paper:**

This paper proposes Knowledge-grounded RL (KGRL) for the purpose of incorporating, reusing, recomposing and generalizing external knowledge in RL tasks. Taking a unified form of external knowledge as an external knowledge policy, this work proposes an actor model that adaptively weighs or activates different external knowledge policies and a learning inner policy. This is realized by a typical self-attention mechanism, with a state-dependent query, inner/external policy-dependent keys. Query, key embeddings and the inner policy are all learned in an end-to-end fashion from standard RL policy signals/gradients. The proposed approach is empirical evaluated in MiniGrid and OpenAI-Robotics benchmark, demonstrating the supriority of the proposed in learning efficiency and generalization ability, given a few pre-defined external knowledge policies.

**Summary Of The Review:**

According to my detailed review above, I think this paper is clearly below the acceptance threshold mainly due to the improper claims on knowledge, the lack of novelty of the proposed method and insufficient experimental evaluation.

---

> ### Author Response · Authors · 2022-11-17
> **Response to Reviewer c6pa (1/3)**
>
> 1. **The major issue is on the central notion of this paper, i.e., Knowledge or KGRL. I think it is improper and kind of misleading to use the notion of knowledge through the paper. After reading the paper, I recommend the authors to use the notion e.g., Policy Reuse/Transfer, which is more accurate.**
>
> Thank you for the suggestion. We would like to refer you to the general response above, “Response to Shared Comments (1/2) - 1,” about the differences between KGRL and policy reuse/transfer.
>
> 2. **The authors emphasize many times in this paper like the claim ‘all knowledge can be arbitrarily recomposed, rearranged, and reused anytime in the learning and inference stages’. However, the concrete form of external knowledge considered is the external knowledge policy.
> In other words, acctually the main content of this work is on how to incorporate, reuse and transfer existing policies in RL. For KGRL at least in my opinion, the key part is how different (or several common) forms of external knowledge are represented, after which how to use the knowledge is to be considered. However, this part is not included in this work and in the experiments, the external knowledge is implemented by the pre-defined policies.**
>
> Thank you for telling us your concern. We would like to point out that (1) our proposed method includes jointly learning representations of different knowledge forms (i.e., their keys/embeddings) and (2) in Section 5.1 “Analysis of Reusing, Recomposing, and Rearranging Knowledge,” we have shown that KGRL can incorporate different forms of knowledge policies (hand-crafted rules and neural networks) into learning.
>
> 3. **Given the external knowledge policies, the proposed method in Section 4 is not novel to me. The self-attention mechanism and the end-to-end update are straightforward.**
>
> Thank you for telling us your concern. We would like to refer you to the general response above, “Response to Shared Comments (1/2) - 3,” about the novelty of KGRL.
>
> 4. **The experiments provide few impressive results, especially when the pre-defined external knowledge policies are strong and the environments are not challenging.**
>
> Thank you for telling us your concern. We would like to refer you to the general response above, “Response to Shared Comments (1/2) - 2”. In short, our predefined external knowledge is non-optimal for most tasks, and the environments we chose are not simple enough to be all solvable using a small number of samples.
>
> 5. **Important related work is missing. For example, Knowledge Guided Policy Network (KoGuN) [1] propose a similar idea, where suboptimal external knowledge is represented by trainable module based on fuzzy logic and then is incorporated into PPO policy network serving as a refine module.**
>
> Thank you for the suggested related work. We have updated our manuscript to include some related work of incorporating external strategies into RL and discuss the differences between our KGRL method and those approaches. In short, our method learns a uniform representation of knowledge in an arbitrary form, hence allowing flexible recomposition/rearrangement/reuse of knowledge.
>
> 6. **The authors claim “current learning-from-demonstration and knowledge-reuse approaches lack the flexibility to rearrange and recompose different demonstrations or knowledge, so they cannot dynamically adapt to environmental changes”. Do the author mean the ability of zero-shot adaptation or generalization? To my knowledge, policy reuse and policy transfer are well-studied domain in this direction. Can the authors justify this claim?**
>
> Thank you for the question. Current LfD and knowledge-reuse methods usually do not allow free add/removal of external guidance during the learning and inference stages, because they couple the knowledge training/selection mechanism with its representation. An example of this coupling can be that a knowledge selection function requires the input of suggested actions. As a result, during a training or inference stage, if the environment changes such that some knowledge is no longer useful or is potentially harmful to the agent, those approaches cannot remove the knowledge.

---

> > ### Comment · Reviewer_c6pa · 2022-11-18
> > **Thanks for careful response!**
> >
> > I appreciate the authors' careful response. Some of my concerns are addressed.
> >
> > For some more feedback:
> > 1) Actually, I am kind of confused with the policy representation (embedding) the authors mentioned and I try to understand it in a similar way to (learnable) word embedding. As to “To the best of our knowledge, there is currently no other work trying to learn a representation of an entire policy and allow flexible recomposition and rearrangement of an arbitrary set of external policies”:
> > for policy representation, there exists a few works on policy representation learning for an entile policy [1-6] (at least since 2018);
> > for flexible recomposition and rearrangement of a set of policies, this work mainly replies on the attention mechanism to deal with varying number of knowledge policies, while as I know, a similar idea of attentive transfer from multiple policy sources is proposed in [7].
> > 2) I see the meaning of “disentanglement” (i.e., the query does not take knowledge representations as input), but “knowledge selection mechanism does not depend on which knowledge set is used” does not make sense to me in a general view. In my opinion, I still view it as a separation rather than a disentanglement.
> > 3) Finally, as I will still recommend the authors to add the additional baseline, i.e., a KGRL variant without state-dependent key, for more convincing experimental results.
> >
> > &nbsp;
> >
> > Reference:
> >
> > [1] Learning Policy Representations in Multiagent Systems. ICML 2018
> >
> > [2] VPE: Variational Policy Embedding for Transfer Reinforcement Learning. ICRA 2019
> >
> > [3] Policy Evaluation Networks. arXiv:2002.11833
> >
> > [4] Goal-Conditioned Generators of Deep Policies. arXiv:2207.01570
> >
> > [5] What about Inputting Policy in Value Function: Policy Representation and Policy-Extended Value Function Approximator. AAAI 2022
> >
> > [6] ASE: large-scale reusable adversarial skill embeddings for physically simulated characters. ACM Trans. Graph. 41(4): 94:1-94:17 (2022)
> >
> > [7] Attend, Adapt and Transfer: Attentive Deep Architecture for Adaptive Transfer from multiple sources in the same domain. ICLR 2017

---

> > > ### Author Response · Authors · 2022-11-18
> > > **Thank you for the reply!**
> > >
> > > Thank you for the comments! Please refer to our responses below.
> > >
> > > 1. **Actually, I am kind of confused with the policy representation (embedding) the authors mentioned and I try to understand it in a similar way to (learnable) word embedding. As to “To the best of our knowledge, there is currently no other work trying to learn a representation of an entire policy and allow flexible recomposition and rearrangement of an arbitrary set of external policies”:**
> > >
> > >     **(1) for policy representation, there exists a few works on policy representation learning for an entile policy [1-6] (at least since 2018);**
> > >
> > >     **(2)for flexible recomposition and rearrangement of a set of policies, this work mainly replies on the attention mechanism to deal with varying number of knowledge policies, while as I know, a similar idea of attentive transfer from multiple policy sources is proposed in [7].**
> > >
> > > First of all, we would like to clarify that what we were trying to say is, “We have not observed previous work trying to **simultaneously** learn a representation of an entire policy and allow flexible recomposition and rearrangement of an arbitrary set of external policies”. After carefully reviewing all the references provided and other previous work, we still could not find work that tries to achieve this.
> > >
> > > Even viewing policy representation and flexible policy recomposition/rearrangement separately, our method is different from previous approaches:
> > >
> > > (1) Policy representation: Previous work **precollects episode data** [1-6], which contains states or states + actions, or **uses all parameters of a policy** [5] to obtain a policy representation by learning from the data or directly concatenating states and actions. Our method **does not require a data precollection process or parameterized policies**, thus policies in different forms can be jointly incorporated.
> > >
> > > (2) Flexible policy recomposition/rearrangement:  In [7], their attention network outputs the weights for **a fixed number of knowledge policies**, while our method calculates a weight by the dot product of a query and a knowledge key, which are independent of each other. Therefore, if a new knowledge policy is added, [7] requires retraining the entire attention network with a new output dimension, and the weights of other knowledge policies will be affected. On the other hand, our method only requires learning the key (embedding) of the new policy, and the weights of other knowledge policies will remain unchanged.
> > >
> > > 2. **I see the meaning of “disentanglement” (i.e., the query does not take knowledge representations as input), but “knowledge selection mechanism does not depend on which knowledge set is used” does not make sense to me in a general view. In my opinion, I still view it as a separation rather than a disentanglement.**
> > >
> > > We use the term “disentangle” because we “disentangle” the weight of choosing a knowledge policy into (1) the representation (key) of the knowledge and (2) the vector (query) that attends to this representation. In this work, the query is independent of the knowledge embeddings (i.e., does not take them as input) and does not contain information about them. Our idea is somewhat similar to “disentangled representation learning” [8,9], e.g., disentangling pose or lightning information from images.
> > >
> > > 3. **Finally, as I will still recommend the authors to add the additional baseline, i.e., a KGRL variant without state-dependent key, for more convincing experimental results.**
> > >
> > > Thank you for the suggestion. We will try to add these experiments in our next paper revision.
> > >
> > > [References]
> > >
> > > [8] Chen, Xi, et al. "Infogan: Interpretable representation learning by information maximizing generative adversarial nets." Advances in neural information processing systems 29 (2016).
> > >
> > > [9] Tran, Luan, Xi Yin, and Xiaoming Liu. "Disentangled representation learning gan for pose-invariant face recognition." Proceedings of the IEEE conference on computer vision and pattern recognition. 2017.

---

> ### Author Response · Authors · 2022-11-17
> **Response to Reviewer c6pa (2/3)**
>
> 7. **The authors claim “These two research directions do not enable incorporating external guidance from different sources into learning, so their transferred knowledge is limited to the one learned among similar environments” and “KGRL allows an agent to follow and reuse an arbitrary set of knowledge obtained from different sources in a new task, and these sources can be very different from the new task”. What do the ‘different sources’ mean? External knowledge in different forms? It is confusing to me since this paper does not address how arbitrary forms of external knowledge are transformed exactly into the external knowledge policies. Instead, the external knowledge policies are given in the setting considered in this work.**
>
> Thank you for the question. Different sources in our work can mean that (1) external knowledge policies come from different processes, such as human-defined, automatically generated, or trained, or (2) external knowledge policies are in different forms, as long as they are state-action mappings.
> In this paper, we propose to learn a uniform representation of a knowledge policy, i.e., its key/embedding, to allow the incorporation of arbitrary external knowledge forms. We also demonstrate this ability in Section 5.1 “Analysis of Reusing, Recomposing, and Rearranging Knowledge,” where hand-crafted and neural-network policies are simultaneously incorporated.
>
> 8. **Using a state-dependent key for the inner policy makes sense to me. However, I recommend the authors to add a KGRL variant without state-dependent key as an additional baseline in the experiments for more complete evaluation and abalation.**
>
> Thank you for the suggestion. The strategy of the inner policy is unknown before training, and its key (the inner key) in the embedding space has more possible positions. Therefore, we design the inner key to be state-dependent to increase its flexibility.
>
> 9. **I also have concern on the claim “we disentangle the decision component (the query) and the knowledge representations (the keys)”. Indeed, they are separated but they are updated in an entangled form regarding the policy gradients.**
>
> Thank you for telling us your concern. This disentanglement means that the knowledge selection mechanism does not depend on which knowledge set is used, e.g., it does not take knowledge representations as input. Hence, an agent should still be able to select a proper knowledge mapping when a different knowledge set is provided. We have updated our manuscript to make this description clearer.
>
> 10. **I think there is a straightforward alternative to the gumble-softmax way to acting with the inner policy and multiple external policies. Concretely, to sample an action for each policy (with reparameterization) and then to mix them according to a softmax distribution of w . Why is this way not considered? Or what is the drawback of this alternative way?**
>
> Thank you for the question. We would like to refer you to Section 4.2 of our updated manuscript, where we explain why we choose to use Gumbel softmax instead of a weighted sum to obtain the final policy in a continuous action space. In short, using a weighted sum in a continuous action space can lose critical information contained in individual actions and lead to worse behaviors.
>
> 11. **Are the external knowledge policies used in the experimenst stochastic or deterministic?**
>
> Thank you for the question. The external knowledge policies are stochastic in all environments. For a discrete action space, an external knowledge policy outputs a probability vector; for a continuous action space, an external knowledge policy outputs a mean action and a fixed standard deviation. We would like to refer you to Appendix A of our updated manuscript, where we add the details of designing those policies.
>
> 12. **It is strange to see the very small error bars in Figure 2a and 2b (in fact, I cannot tell if there is a error bar in Figure 2b).**
>
> Thank you for telling us your concern. All experiments in this paper are run with 5 different random seeds. The reason why Figure 2a and 2b shows little variance can be that the tasks are simple, so there is not much difference among the training processes.

---

> ### Author Response · Authors · 2022-11-17
> **Response to Reviewer c6pa (3/3)**
>
> 13. **In Figure 2e, why do PPO-KGRL (KG3) and PPO-KGRL (KG1+KG2) perform worse than PPO while PPO-KGRL (all) outperforms PPO?**
>
> Thank you for the question. In a small environment such as DoorKey-5x5, (1) indirect guidance, i.e., KG1+KG2, which does not directly lead an agent to a reward, or (2) naive guidance, i.e., KG3, which ignores the intermediate steps necessary to reach the goal, may be more challenging to benefit the agent in a short amount of time due to extra learnable parameters in our actor model.
>
> 14. **In Section 5.2, it seems that KG1 is already optimal in Reach task and KG1 + KG2 is optimal in Pick-and-Place. Can the authors provide more explanation on this point? Similarly, KG1 + KG2 + KG3 is also optimal in 5x5 environments since the view range is also 5x5.**
>
> Thank you for the question. We would like to refer you to the results of BC in our updated manuscript (Section 5). The external knowledge policies are non-optimal in most tasks. Also, KG1+KG2+KG3 in MiniGrid 5x5 environments is not optimal because a state is in the first-person view, which means that an agent can only see what is in front of it.
>
> 15. **The authors mention “indicating that a KGRL agent learns to not only follow the guidance but also mimic its strategy. This imitation process allows an agent to develop an inner policy that outperforms external strategies”. Can the authors provide more explanation on how such imitation happen under KGRL and why imitation can be better than the external policies?**
>
> Thank you for the question. During the learning process, an agent can observe actions provided by the external knowledge policies and their corresponding rewards. These learning signals will be back-propagated to all learnable components, including the inner policy, during the end-to-end learning process. If those actions lead to higher rewards, the inner policy can learn to imitate them. Moreover, the inner policy can generate its own actions, which can be different from that of the external knowledge policies and encourages exploration in the environment. With these two sources of action data, the inner policy may outperform all external policies after being trained by an RL algorithm.
>
> 16. **Some key details of the pre-defined external knowledge policies are missing (e.g., ϵ in Section 5.2). The source codes are not provided.**
>
> Thank you for indicating the clarity issue. We would like to refer you to Appendix A of our updated manuscript, where we provide the experimental details and code for defining the external knowledge. We are cleaning up our code and will publish it soon.

---

### Author Response · Authors · 2022-11-17
**General Response - Manuscript Update**

We thank all the reviewers for their precious time and feedback.

We have updated the manuscript according to the comments. The updated contents are marked as blue in the manuscript. We summarize the changes compared to the previous version as follows:

1. **Related Work**: We added some related work about incorporating external strategies into RL.
2. **KGRL Problem Formulation**: We had stated that each knowledge $g$ is a mapping between a state and an action (not text) in the original version. To prevent misunderstanding, we edit the manuscript to clarify more that this mapping can be in any form (hand-crafted rules, fuzzy logic, or neural networks) as long as it outputs an action given a state.
3. **Proposed Approach**: We explain our idea of designing the proposed actor model in more detail. In addition, we answer the following raised questions in the updated version:

    (1) Why is a knowledge key independent from a state or action?

    (2) What do we mean by “disentangle the selection mechanism from a knowledge set”?

    (3) Why do we use the Gumbel-softmax trick instead of a weighted sum for all action candidates in a continuous action space to obtain a final policy?

    (4) How are all components in the proposed actor model learned?

4. **Experiments**: We compare our KGRL method with two additional baselines, behavior cloning (BC) and RL+BC [1]. The BC results demonstrate that the rules we used cannot succeed in all environments except OpenAI-Robotics Reach. This shows that our KGRL agent can learn effectively by referring to sub-optimal guidance. Moreover, our KGRL approach learns more efficiently than RL+BC, which is a popular way to incorporate external guidance into RL.
5. **Appendix**: We add more experimental details and the implementation of the predefined rules here.

[1] Nair, Ashvin, et al. "Overcoming exploration in reinforcement learning with demonstrations." 2018 IEEE international conference on robotics and automation (ICRA). IEEE, 2018.

---

### Author Response · Authors · 2022-11-17
**General Response - Response to Shared Comments (1/2)**

We would like to first address some concerns shared by some reviewers here.

1. **(Reviewer c6pa and uBBQ) The term “knowledge” is misleading. Sub-policy reuse/transfer is more accurate.**

Thank you for the suggestion. This paper focuses on “how to recompose, rearrange, and reuse an arbitrary set of external policies to improve efficiency in RL” instead of learning a reusable policy. In other words, the key components of our proposed method are the policy representations (embeddings) and the attention mechanism, which enable flexible combinations of policies. Policy reuse/transfer in previous work, on the other hand, focuses on learning transferable policies and reusing them (no external one is considered). This can be achieved by our method, as explicitly demonstrated in Section 5.1 “Analysis of Reusing, Recomposing, and Rearranging Knowledge,” but is not the main focus of this work.

Our definition of “knowledge” in this work is “a strategy when addressing a task,” which we more specifically defined as a mapping between a state and an action. This is in line with the definition of “knowledge” in previous literature on knowledge transfer/integration in RL [2-7].

2. **(Reviewer c6pa, uBBQ, and ce5j) The pre-defined external knowledge policies are strong, and the environments are not challenging. / The paper does nothing to convince the reader that the proposed method would go beyond toy tasks where it is easy to provide pre-baked policies. / In practice, the pre-baked policies provided to the agents in the paper are very good examples of skills and options. / The knowledge in the experiments is simply commands or high-level options. The authors only construct instruction-following tasks rather than knowledge-incorporating tasks.**

Thank you for telling us your concern. We would like to point out that (1) we intentionally designed the external knowledge to be very simple, (2) the knowledge is not optimal in most environments, and (3) the MiniGrid and OpenAI robotic environments we chose are not all toy environments that can be solved by RL using a reasonable number of samples.  Below we elaborate more on each point:

(1) We want to demonstrate that even a naive strategy can guide an agent to complete a task more efficiently. For example, KG3 in Section 5.1 and KG1 and KG2 in Section 5.2 tell an agent, “If you want to reach a goal, just go straightly in the direction of it.” This is not optimal for many navigation, manipulation, and motion planning tasks but is a good heuristic when solving these tasks.

(2) The knowledge we use in this work is only optimal for Reach. Please refer to the results of BC in Section 5 of the updated manuscript. We also demonstrate in Figure 3 that adding redundant knowledge will not affect the results and can help explore an environment. This shows that knowledge in KGRL is not restricted to useful skills or options.

(3) These tasks have different difficulty levels, including simple ones and ones that widely-used RL algorithms cannot learn, so they can better demonstrate our method's efficiency, generalizability, compositionality, and incrementality [8].

3. **(Reviewer c6pa and ce5j) Given the external knowledge policies, the proposed method in Section 4 is not novel to me. The self-attention mechanism and the end-to-end update are straightforward. / The proposed method is not novel. Attentional architecture is the de-facto choice for building instruction-following agents.**

Thank you for telling us your concern. To the best of our knowledge, there is currently no other work trying to learn a representation of an entire policy and allow flexible recomposition and rearrangement of an arbitrary set of external policies. Our method is straightforward, but it is also novel, as no one has proposed it to incorporate demonstrations in RL, and effective, as shown in our experiments and analyses. In addition, this work is not about building instruction-following agents but about incorporating external guidance to help an agent learn more efficiently.

---

### Author Response · Authors · 2022-11-17
**General Response - Response to Shared Comments (2/2)**

4. **(Reviewer quir and 4vDB) Many important details of the proposed method are missing, making many parts of this work difficult to follow. / The implementation details are very general. I don’t think it would be easy to reproduce the results.**

Thank you for indicating the clarity issue. We have updated our manuscript to include (1) more explanation about our problem formulation and proposed method in Section 3 and 4 and (2) more experimental details in Section 5 and Appendix A. We also included our code for defining knowledge policies in Appendix A. We are cleaning up our code and will publish it soon.

5. **(Reviewer quir and 4vDB) Does the term “new knowledge” simply refer to the stringing together of old knowledge? If not, could you please give some experimental examples? / It’s not clear what the internal policy is learning. It seems to be that the internal policy is learning to explore the environment and relies on external knowledge for task execution.**

Thank you for indicating the clarity issue. The internal policy can learn its own strategy to partially or fully complete a task. Some examples can be found in Figure 3, Figure 6, and Table 2 in the updated manuscript. In Figures 3 and 6, since the external knowledge is not optimal in all environments (as indicated by the results of BC), the inner policy has to develop its own knowledge to solve a task. In Table 2, an inner policy itself can solve Pick-And-Place, while KG1+KG2 can hardly solve it.

6. **(Reviewer 4vDB and ce5j) The paper misses related work in text-based RL. Text-based RL benchmarks are good candidates for this work. / The authors only construct instruction-following tasks rather than knowledge-incorporating tasks.**

Thank you for telling us your concern. This work is not related to text-based RL or learning instruction-following agents. Their goals are completely different from that of KGRL. The former aims to “understand and follow” the given language instructions, while KGRL aims to learn a task more efficiently by “referring to” external guidance/demonstrations, and a KGRL agent is not required to follow external guidance. In addition, the form of knowledge in KGRL is a state-action mapping (policy) instead of text or language.

[References]

[2] Parisotto, Emilio, Jimmy Lei Ba, and Ruslan Salakhutdinov. "Actor-mimic: Deep multitask and transfer reinforcement learning." arXiv preprint arXiv:1511.06342 (2015).

[3] Rusu, Andrei A., et al. "Policy distillation." arXiv preprint arXiv:1511.06295 (2015).

[4] Yin, Haiyan, and Sinno Jialin Pan. "Knowledge transfer for deep reinforcement learning with hierarchical experience replay." Thirty-First AAAI conference on artificial intelligence. 2017.

[5] Xu, Zhiyuan, et al. "Knowledge transfer in multi-task deep reinforcement learning for continuous control." Advances in Neural Information Processing Systems 33 (2020): 15146-15155.

[6] Tao, Yunzhe, et al. "Repaint: Knowledge transfer in deep reinforcement learning." International Conference on Machine Learning. PMLR, 2021.

[7] Zhang, Peng, et al. "KoGuN: accelerating deep reinforcement learning via integrating human suboptimal knowledge." arXiv preprint arXiv:2002.07418 (2020).

[8] Kaelbling, Leslie Pack. "The foundation of efficient robot learning." Science 369.6506 (2020): 915-916.

---

### Decision · Program_Chairs · 2023-01-20

**Decision:**

Reject

**Justification For Why Not Higher Score:**

The experiments in this paper are limited due to the simplicity of the task and environments. There was a consensus about this with all reviewers. The use of toy environments like MiniGrid and simple control tasks are well suitable to establish the efficacy of the approach but does not provide any insights or path for scaling. They are good to demonstrate proof of concept but insufficient to establish and build upon for more complicated and real world problem domains. The level of compositionality required to solve the tasks should be expanded to make the paper and experiments stronger.

The paper also relies heavily on manually crafted policies. It is not clear how to learn or extract them from experience or learning from other tasks. The paper should have evaluated more robust and general policies to better assess the proposed method's capabilities and limitations.

**Justification For Why Not Lower Score:**

N/A

**Metareview: Summary, Strengths And Weaknesses:**

This paper studies the problem of integrating knowledge bases with reinforcement learning. The problem is formulated as doing RL by learning to index into a knowledge base of pre-trained policies (+ an online learned policy), using an attention based mechanism. The approach is validated on MiniGrid and robotics environments.

The approach is clearly written and there was no confusion about the proposed approach. However, there was a strong consensus that the environments considered were too simplistic to establish the generality of this approach. There are a lot of assumptions made in this paper about how the policies are hand constructed and the dimensionality of the knowledge base needed for generality. Without scaling up the approach to more realistic environments, it will be difficult to build upon this work and clearly establish claims.